# GDA: Grammar-based Data Augmentation for Text Classification using Slot Information

**Joonghyuk Hahn[1]**    **Hyunjoon Cheon[1]**    **Elizabeth Orwig[1]**
**Su-Hyeon Kim[2]**    **Sang-Ki Ko[3]**    **Yo-Sub Han[1]**

[1]Yonsei University    [2]Kangwon National University    [3]University of Seoul

{greghahn,hyunjooncheon,orwig,emmous}@yonsei.ac.kr
tngus987207@kangwon.ac.kr    sangkiko@uos.ac.kr

## Abstract

Recent studies propose various data augmentation approaches to resolve the low-resource problem in natural language processing tasks. Data augmentation is a successful solution to this problem and recent strategies give variation on sentence structures to boost performance. However, these approaches can potentially lead to semantic errors and produce semantically noisy data due to the unregulated variation of sentence structures. In an effort to combat these semantic errors, we leverage slot information, the representation of the context of keywords from a sentence, and form a data augmentation strategy which we propose, called GDA. Our strategy employs algorithms that construct and manipulate rules of context-aware grammar, utilizing this slot information. The algorithms extract recurrent patterns by distinguishing words with slots and form the "rules of grammar"—a set of injective relations between a sentence's semantics and its syntactical structure—to augment the dataset. The augmentation is done in an automated manner with the constructed rules and thus, GDA is explainable and reliable without any human intervention. We evaluate GDA with state-of-the-art data augmentation techniques, including those using pre-trained language models, and the result illustrates that GDA outperforms all other data augmentation methods by 19.38%. Extensive experiments show that GDA is an effective data augmentation strategy that incorporates word semantics for more accurate and diverse data.

## 1   Introduction

Text classification is a popular application of deep learning (Minaee et al., 2021), and BERT-based language models achieve a promising performance for the task (Devlin et al., 2018; Liu et al., 2019; Lan et al., 2019). These popular language models, however, often exhibit a critical performance drop when the labeled data are insufficient.

| Methods | |
|---|---|
| Methods | **Original**: Add a song by Bruno Mars to my playlist Bruno Mars Hits. |
| EDA | Add a song Bruno Mars by Uptown Funk to my playlist. |
| | Add a song by Mars Bruno to my playlist Bruno Hits Mars. |
| SSMBA | Add my playlist Bruno Mars Hits to a song by Bruno Mars. |
| | Add a song to my playlist Bruno Mars Hits by Bruno Mars. |
| ALP | Add a song by Bruno Mars to my Bruno Mars Hits playlist. |
| | Add a song by Bruno Mars to my playlist Bruno Marses Hits. |
| GDA | Please add Bruno Mars to my playlist Bruno Mars Hits. |
| | Insert a song of Bruno Mars to Bruno Mars Hits. |

Table 1: Example sentences generated by GDA and state-of-the-art data augmentation methods. Note the logical errors generated by EDA, SSMBA, and ALP as opposed to GDA which generates contextually equivalent sentences with variations.

Language models often fail to learn the data distribution if the (sampled) dataset is too small to characterize the population (low-resource problem). Recent studies propose many techniques that alleviate the low-resource problem, such as data augmentation—a broadly accepted technique in the field of NLP. Many data augmentation approaches such as EDA (Wei and Zou, 2019) and ALP (Kim et al., 2022) focus on syntactic variation to produce diverse data and have shown a definitive performance improvement. However, these strategies are not always reliable because they can lead to semantic errors as the variation might negatively impact the meaning of a sentence. We present some examples of sentences with logical errors in Table 1. As a result, the augmented sentences, which are potentially nonsensical, may be harmful to a model degrading classifier performance. On the other hand, UDA (Xie et al., 2020) also shows definitive performance improvement, but employs back-translation and word replacement strategies in order to prevent semantic errors from occurring, limiting the production of syntactically versatile augmented sentences.

In text classification, a data sample pairs a nat-

ural language sentence with its class, and a model learns to determine the classes from given sentences. Several models utilize relations between the words to understand the given sentences, such as entities, part-of-speech (POS), and role semantics (Wu et al., 2018; Sood et al., 2020; Zhang et al., 2022). We can use this information to augment the training data and improve the performance of language models.

Recent studies represent each sentence with a slot sequence in an effort to enhance sentence classification (Goo et al., 2018; Wu et al., 2020; Qin et al., 2021). These studies, however, show limitations on utilizing slot information because they only apply this information during the training phase of the models. We leverage slot information to enrich the diversity of generated sentences in data augmentation.

Given a sentence $S = w_1 w_2 \cdots w_n$, there is a sequence of slots $s = s_1 s_2 \cdots s_m$ of $S$, which contains contextual information of words in $S$ where $m \leq n$, and there is a class-label $I$ of $S$, in which $I$ represents the perceived intention of the sentence. See Figure 1 for an example.

| | | | | | | | |
|---|---|---|---|---|---|---|---|
| $S$ = | Please | add | *Iris DeMent* | to | my | playlist | *this is Selena* |
| $s$ = | – | – | artist | – | – | – | playlist |
| $I$ = | AddToPlaylist | | | | | | |

Figure 1: Example of a sentence $S$, its slot sequence $s$ and class $I$. Words labeled by '–' denote the words with no corresponding contextual information.

While data augmentation in recent studies depends heavily on words present in the sentences, we focus more on the implicit properties of the words. For instance in Figure 1, the slot 'artist' indicates that '*Iris DeMent*' describes a name of an 'artist'. The other underlined phrase '*this is Selena*' could be understood as an individual sentence. However, the slot 'playlist' associated with the phrase implies that the phrase indicates a name of playlist, rather than a sentence by itself. These properties aid the model in comprehending such sentences precisely and act as a crucial component for interpreting the sentences. Thus, we utilize these implicit properties to gain more appropriate data for each class.

Slots work by partitioning a sentence into meaningful word phrases. Hahn et al. (2021) utilize the characteristics of slots in semi-supervised learning for pseudo-labeling procedures. Their work infers a set of rules by substituting word phrases

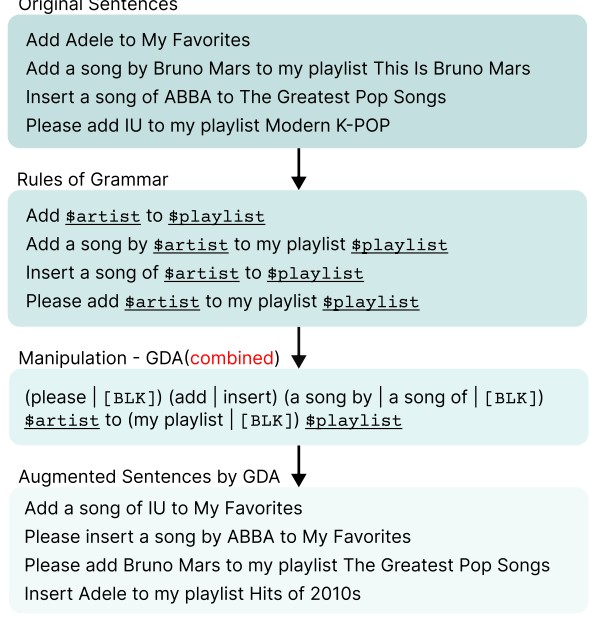

Figure 2: An overview of GDA. It constructs RoGs by distinguishing keywords with slot information. We employ a manipulation for rules in more broad forms and then, generate new sentences. The symbols preceding a dollar sign ($) are substitution variables for slots.

with symbols representing slots. After analysis, the authors conclude the approach is effective for low-resource problems.

In a similar vein, we propose grammar-based data augmentation (GDA) using slots. We first construct rules of grammar (RoGs) by substituting keywords in the sentences with their coinciding slots. We then apply heuristics inspired by formal language to generalize these RoGs. Such manipulations provide variations in sentence structures and augment syntactically propitious sentences. We then produce a variety of sentences from the RoGs and augment them to the training data. Figure 2 describes the overview of GDA. Our definition of GDA consists of three parts: RoG construction, rule manipulation, and sentence generation.

GDA is a powerful data augmentation method when conducted on sentences with slots. It shows a maximum of 19.38% improvement in terms of prediction accuracy in the 5-shot experiment of SST-2. We also expand our method to more general types of datasets where slots are not initially available. With an intuition that named entities contain contextual information of the sentences (Mohit, 2014), we conduct named entity recognition (NER) on the sentences and infer RoGs by substituting words with their named entities. The experiment compares state-of-the-art data augmentation techniques

to GDA and demonstrates that GDA is a successful implementation on typical text classification.

## 2 Related works

Data augmentation is a solution to the data scarcity problem which works by generating artificial data from the original dataset, with the key objective producing accurate and diverse data. Traditional data augmentations involve synonym replacements or random word insertion, deletion, or substitution. More advanced augmentation techniques, such as symbolic augmentation, utilize alternative representations of natural language sentences (Zhang et al., 2020; Wei and Zou, 2019). For example, EDA (Wei and Zou, 2019) applies predefined operations (rules) to modify sentences and Coulombe (2018) proposes utilizing regular expressions to filter and clean noisy data while augmenting it.

Another line of research in data augmentation focuses on conditional generation using pre-trained language models—the results of which have been promising, especially given their significant performance improvements (Devlin et al., 2018; Raffel et al., 2020). Large pre-trained language models like GPT-3 (Brown et al., 2020) have gained popularity in data augmentation. However, the resource requirements for these models as well as the process of fine-tuning such large models for specific datasets is often infeasible and poses a challenge for individual users and researchers.

Recent studies explore the use of word properties from given sentences, such as slot information in task-specific datasets, like ATIS and Snips, or named entities. In our work, we utilize this contextual information to overcome the challenges of employing large language models (LLMs) and we then propose a rule-based data augmentation approach based on context-free grammar.

## 3 Methodologies

GDA works in three phases: inference, manipulation of the RoGs, and sampling sentences over the rules. Given a set $S$ of sentences, we first infer the set $R$ of RoGs from $S$. In the inference phase, GDA identifies keywords—word phrases to which a slot is attached—of sentences in $S$ and replaces these phrases with a variable representing the slots, thus contextually equivalent phrases are replaced with the same variable. For instance, 'Bruno Mars' and 'Adele' are replaced by '$artist'. Then in the manipulation phase, GDA combines and modi-

fies each set of the inferred rules $R$ to form a larger set of rules $R'$ that represents common structures of sentences. We present a detailed overview of the manipulation phase of GDA in Figure 3. In this phase, we mix the rules carefully while maintaining their semantics. GDA then generates new sentences represented by the rules $R'$.

### 3.1 Rule inference

An initial ruleset $R$ for given sentences is *context-free grammar* and is constructed by replacing every keyword with a variable for its slot. Context-free grammar $G = (V, T, P, V_0)$ is generative grammar that consists of two disjoint sets, $V$ and $T$, of variables and terminals respectively, a set $P \subseteq V \times (V \cup T)^*$ of production rules and a start variable $V_0 \in V$. We refer to the items in $V \cup T$ by 'tokens'. A production $p \in P$ pairs a variable $A$ and a sequence $X$ of tokens, which we denote $p = A \to X$. We refer to the variable and the sequence in a production $p$ as a substitution variable and a substitution sequence respectively. We often denote several productions $A \to X_1$, $A \to X_2, \ldots A \to X_n$ having the same substitution variable in an alternation sequence (or, alternation) $A \to X_1|X_2|\cdots|X_n$. We also use parentheses to denote a substitution sequence in single production instead of in separated productions for conciseness. For instance, '$A \to (add|insert)$ *a song*' is a short notation of these three productions: '$A \to A'$ *a song*', '$A' \to add$', and '$A' \to insert$'. A ruleset $R = (V, T, P, V_0)$ generates a sentence by starting from $V_0$ and replaces a substitution variable with one of its substitution sequences in $P$ until there are no variables left.

Given a sentence $S = w_1 w_2 \cdots w_n$ in class $I$ with its slot sequence $\boldsymbol{s} = s_1 s_2 \cdots s_m$, for every $w_i$ where its corresponding slot $s_j$ contains contextual information (i.e., $s_j$ is not $-$), we replace such $w_i$'s with its slot $s_j$. Then, we pair the class label $I$ and this modified sentence $S'$ into a production. Finally, we pair the start variable $V_0$ and every class label $I$. Thus, we can build a context-free grammar $G = (V, T, P, V_0)$ where $V$ is a set of slots, class labels, and other variables and $T$ is a set of words.

These initial rules are effective but scarcely produce sentences with high syntactic variation because they maintain the structure of underlying sentences. This limitation of the initial rules, similar to the limitation of self-training, results in possibly

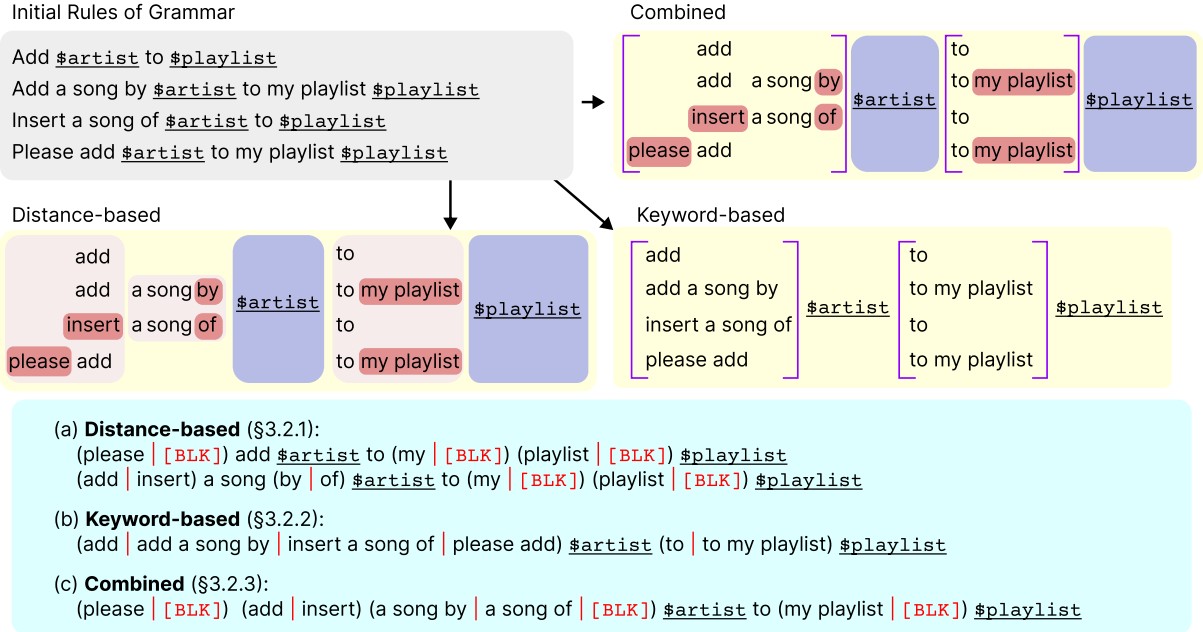

Figure 3: An overview of GDA manipulation methods. Details of manipulation strategies are in the yellow box and the resulting rules by manipulations are in the blue box. `[BLK]` denotes an empty word.

as many rules as there are sentences in the class. This limitation potentially hinders the diversity of generated rules and thus the generation of varying sentences. For example, two rules '*add* `$song` *to playlist*' and '*insert the song* `$song` *to playlist*' ask to add the song denoted by the slot '`$song`' to some playlist, but the rules fail to generate a sentence like $S_1 =$ 'add the song *dynamite* to playlist' as the word phrase '*add the song*' is not recognized by both rules. However, we can make these rules generate $S_1$ by allowing the second rule to have '*add*' in place of '*insert*' or by allowing the first rule to generate '*the song*' between '*add*' and '`$song`'. On the other hand, if we allow the second rule to generate '*remove*' in place of '*insert*', the semantics of the generated sentence diverges from the class label (and the initial sentences). Thus it is important to preserve underlying semantics and grammar structure while transforming the initial rules to represent various sentences.

## 3.2   Rule manipulation

With the initially inferred rules, we propose techniques based on formal structures to extend the rules' coverage. We first introduce a distance-based rule manipulation algorithm which utilizes edit-distance to extend the structure of rules. We then introduce a keyword-based rule manipulation algorithm which employs slot information to generate more contextually diverse sentences. The following

sections briefly explain how these two rule manipulation algorithms work.

### 3.2.1   Distance-based rule manipulation

Edit-distance $d(A, B)$ is a dissimilarity metric over two substitution sequences $A$ and $B$, defined as the minimum cost of editing $A$ into $B$, word-by-word. We use three types of rules to edit sequences: inserting, deleting, or replacing a word. They have a unit cost and are often referred to as edit operations. For instance, words such as '*IU*' or '*Michael Jackson*' qualify for word-to-word edits but a slot, such as '`$artist`', does not. Thus we expect that distance-based manipulation preserves the semantics, since $d(A, B)$ anchors the sentence around the slots (the main keywords) and modifies the surrounding words. The syntactic structure is preserved since distance-based manipulation does not reorder the structure of the rules.

One concern with the distance function $d(A, B)$ is that it *counts* the edit operations, which is often proportional to the length of $A$ and $B$, where the length of a rule is defined as the number of tokens in the rule. This leads to the false implication that only the number of different words is important. For example, if there exist two rules with 7 different words and a length of 50 each, as well as two rules with 5 different words and a length of 10 each, the dissimilarity of these rules would be misinterpreted when compared by counting edit

operations. Thus we use normalized distance function $d'(A, B) = d(A, B)/\max\{|A|, |B|\}$ instead of $d(A, B)$, where $|A|$ and $|B|$ are the lengths of $A$ and $B$ respectively.

---

**Algorithm 1** Distance-based manipulation. The inputs $R$ and $\Theta$ in DIST-MANIPULATION$(R, \Theta)$ are a set of rules and an upper threshold for normalized edit-distance, respectively.

---

```
 1: function EDIT-SEQUENCE(A, B)
 2:     return a sequence ((a₁, b₁), ..., (aₙ, bₙ)) of pairs
        where A = a₁a₂···aₙ, B = b₁b₂···bₙ, and
        ∑ⁿᵢ₌₁ c(aᵢ, bᵢ) = d(A, B).
 3: end function
 4: function DIST-CLUSTER(R, Θ ∈ (0, 1])
 5:     i ← 1
 6:     while R ≠ ∅ do
 7:         Choose rᵢ ∈ R at random.
 8:         Cᵢ ← {x ∈ R | d'(rᵢ, x) ≤ Θ and,
                   rᵢ and x have the same class label}
 9:         R ← R \ Cᵢ
10:         i ← i + 1
11:     end while
12:     return {C₁, C₂, ..., C_N}
13: end function
14: function DIST-MERGE(Cᵢ = {rᵢ, r_{i,1}, r_{i,2}, ..., r_{i,n}})
15:     r ← rᵢ
16:     for j ∈ [1, n] do
17:         seq ← EDIT-SEQUENCE(rᵢ, r_{i,j})
18:         r' ← [BLK]
19:         for (x, y) ∈ seq do
20:             if x = y then
21:                 r' ← r' · x          ▷ append the item as is
22:             else
23:                 r' ← r' · (x|y)      ▷ append an alternation
24:             end if
25:         end for
26:         r ← r'
27:     end for
28:     return r
29: end function
30: function DIST-MANIPULATION(R, Θ ∈ (0, 1])
31:     C₁, C₂, ..., Cₙ ← DIST-CLUSTER(R, Θ)
32:     R' ← ⋃ⁿᵢ₌₁ DIST-MERGE(Cᵢ)
33:     return R'
34: end function
```

---

Algorithm 1 shows a pseudocode for the distance-based manipulation method. The main function is DIST-MANIPULATION$(R, \Theta)$ (line 30) that takes two inputs, the set $R$ of all rules and a threshold $\Theta \in (0, 1]$ on normalized edit-distance. DIST-MANIPULATION calls two functions DIST-CLUSTER and DIST-MERGE.

DIST-CLUSTER (line 4) partitions the given rules $R$ into several clusters $\{C_i\}_{1 \leq i \leq N}$. Based on a randomly chosen rule $r_i$, DIST-CLUSTER builds a cluster $C_i$ by collecting rules whose minimum normalized edit-distance from $r_i$ is at most $\Theta$. DIST-MERGE (line 14) then analyzes the editing sequence of the representative rule $r_i$ and the other rules $r_{i,j}$

$(1 \leq j \leq n)$ to build a merged rule $r$.

In DIST-MERGE, $(x, y)$ denotes an alignment of two items $x$ and $y$. When an alignment pair consists of the same item, the algorithm uses the item $x$. On the other hand, we use the alternation of two target items $(x|y)$ so that the inference step can choose one of the possible target items. For instance in Figure 3(a), '*add*' and '*insert*' are aligned and the alternation of two words '*add|insert*' is written where '*add*' and '*insert*' appear.

### 3.2.2 Keyword-based manipulation

The second approach gives priority to the semantics of given RoGs. Since keywords imply the main intention of a sentence, we hypothesize that inspecting keywords in every rule can aid in categorizing rules with similar contexts. Under this assumption, we use the slots as keywords for each sentence. Keyword-based manipulation replaces the words and expands the representative power of rules to contain several new sentences of different forms.

Algorithm 2 describes the keyword-based manipulation and Figure 3(b) shows a running example of keyword manipulation on four rules.

---

**Algorithm 2** Keyword-based manipulation. $R$ is a set of RoGs.

---

```
1: function KEY-MANIPULATION(R)
2:     P ← partition R by the slots in rules
3:     for P ∈ P do
4:         Extract maximal word sequences in P
5:         Construct a new rule r by adding alternations
               of extracted sequences and slots
6:     end for
7:     return r
8: end function
```

---

Note that the example rules in Figure 3(b) show different results from that of distance-based manipulation in Figure 3(a): Keyword-based manipulation merges a set of sequence fragments, while distance-based manipulation generates alternations composed only of words.

### 3.2.3 Combination strategy

We also present a third approach, a hybrid strategy consisting of both the aforementioned manipulation methods in order to generalize the representative power of RoGs further. This approach constructs rules that have more syntactic and semantic variation by combining the advantages of both manipulation strategies. For example, applying keyword-based manipulation followed by the distance-based manipulation restricts the edit-distance to work only in the alternations. This

reduces the syntactic errors of the rules by merging the word phrases grouped in an alternation as well as individual words.

## 3.3 Sentence generation

We generate sentences from the manipulated rules $r = x_1 x_2 \cdots x_n$ by substituting slots with their representing words or word phrases and varying the surrounding words. An alternation or a slot often represents several word phrases. Thus, we can generate several distinct sentences. Algorithm 3 illustrates the generation procedure.

---

**Algorithm 3** Rule-based sampling algorithm for augmentation. All selections are at random.

---

1: **function** RETRIEVE-WORD($x$)
2:     **return** a word or a word phrase having slot $x$.
3: **end function**
4: **function** GENERATE($r = x_1 x_2 \cdots x_n$)
5:     **for** $x_i \in r$ **do**
6:         **if** $x_i$ is a slot ($xxx) **then**
7:             $w_i \leftarrow$ RETRIEVE-WORD($x_i$)
8:         **else if** $x_i$ is an alternation $(a_1|a_2|\cdots)$ **then**
9:             Choose $a_j$
10:             $w_i \leftarrow$ GENERATE($a_j$)
11:         **else**
12:             $w_i \leftarrow x_i$       ▷ $x$ should be a word
13:         **end if**
14:     **end for**
15:     **return** a sentence $w_1 w_2 \cdots w_n$.
16: **end function**

---

## 4 Experiment settings

Following the few-shot settings of recent data augmentation research (Lin et al., 2023), we train the baseline models on 5- and 10-shot train sets sampled from each target dataset to simulate a few-shot environment. We generate 500 sentences per class on each combination strategy. We also measure the prediction performance on the original, full dataset to estimate the best performance of the model on each dataset. We present the performance of the full datasets that have slots in Appendix B.

### 4.1 Dataset

We target two types of datasets: those which contain slot information and those which do not. For datasets without slot information, we label entities for each word in substitution for slots using a pre-trained BERT in Stanford CoreNLP (Manning et al., 2014) for NER task. Table 2 summarizes the details of datasets used in the experiment. A set of classes of a dataset is a set of distinct labels of each sentence in the dataset.

| Dataset (without slots) | #Class | #Train | #Valid | #Test |
|---|---|---|---|---|
| AGNews | 4 | 120,000 | - | 7,600 |
| IMDB | 2 | 25,000 | - | 25,000 |
| SST-2 | 2 | 67,349 | 872 | - |
| Yahoo | 10 | 1,400,000 | - | 60,000 |

| Dataset (with slots) | #Class | #Train | #Valid | #Test |
|---|---|---|---|---|
| ATIS | 17 | 4,502 | 500 | 893 |
| Snips | 7 | 13,084 | 700 | 700 |
| FB-MTO | 12 | 30,521 | 4,181 | 8,621 |
| MultiDoGO (6 subcategories) | | | | |
|   - Airline | 11 | 15,438 | 2,053 | 4,063 |
|   - Fastfood | 14 | 15,391 | 2,101 | 4,485 |
|   - Finance | 19 | 14,493 | 2,212 | 4,421 |
|   - Insurance | 10 | 12,234 | 2,039 | 4,102 |
|   - Media | 16 | 13,551 | 1,747 | 3,488 |
|   - Software | 16 | 13,084 | 1,840 | 3,858 |

Table 2: Summary of original datasets.

The three datasets: AGNews, IMDB, and Yahoo! answers (Yahoo) do not contain a validation set. We therefore split the initial train set into train and validation sets which are disjoint to each other. On the other hand, the SST-2 dataset does not contain a labeled test set, and due to this we use the validation set in its stead, splitting the train set to generate a replacement validation set.

### 4.2 Baseline

We use SlotRefine (Wu et al., 2020) and a co-interactive transformer (Qin et al., 2021) as two baseline models to evaluate GDA, as well as three state-of-the-art augmentation strategies: EDA (Wei and Zou, 2019), SSMBA (Ng et al., 2020), and ALP (Kim et al., 2022). These models utilize slot information for sentence class prediction and thus we use them as a basis for comparison. Also for proper analysis on the relative effectiveness compared with LLMs, we employ Bard (Thoppilan et al., 2022) and fine-tune BERT (Devlin et al., 2018) on each dataset for data augmentation.

### 4.3 Evaluation metrics

For the datasets without slots, we simply report the accuracy score. However, for the datasets with slots, we evaluate the effectiveness of our strategy with the average of $F_1$ scores in order to decouple the score for each class label from the number of sentences for each class in the test dataset. We report scores of all combined strategies for each dataset in Appendix B.

## 5 Results and discussion

We employ GDA to augment datasets from Section 4.1 and use the baseline classifiers from Sec-

|  | AGNews | | IMDB | | SST-2 | | Yahoo | |
|---|---|---|---|---|---|---|---|---|
|  | 5 | 10 | 5 | 10 | 5 | 10 | 5 | 10 |
| EDA | 78.89 | 80.72 | 60.32 | 69.80 | 56.22 | 53.96 | 55.49 | 63.12 |
| SSMBA | 78.65 | 84.68 | 66.43 | 63.36 | 56.34 | 59.05 | 53.17 | 61.50 |
| ALP | 82.30 | **86.18** | 67.05 | 71.29 | 63.40 | 69.72 | **55.19** | **64.16** |
| GDA: without manipulation | 82.66 | 82.93 | 55.21 | **76.60** | 69.95 | 75.57 | 47.19 | 52.00 |
| GDA: combined manipulation | **83.13** | 84.22 | **67.32** | **76.60** | **75.69** | **78.78** | 49.32 | 53.06 |

Table 3: Performance comparison of GDA and other baseline data augmentation strategies (EDA, SSMBA, and ALP). The accuracy (%) on each setting is reported and bold scores are the highest scores on each setting. The results on EDA, SSMBA, and ALP are from Kim et al. (2022).

tion 4.2 to evaluate the performance of augmentation. We illustrate the effectiveness of GDA in Tables 3 and 4. We report scores of all combined strategies in Appendices A and B.

## 5.1 Comparison of data augmentation strategies

Table 3 compares GDA with the baseline data augmentation models: EDA, SSMBA, and ALP. By utilizing named entities as slot information, GDA achieves compelling performance compared to baseline augmentation models.

Among the baselines from Section 4.2, ALP is the state-of-the-art model which utilizes POS tagging for data augmentation (Kim et al., 2022). It uses a formal conception—probabilistic context-free grammar (PCFG), the rules of which introduce hierarchical structure variation while preserving the context of the initial data. This syntactic variation leads to decent improvements in the performance of text classification. On the other hand, since PCFG reflects the distribution of syntactic variation in the original dataset, it is hard to provide sentences with novel syntactic forms. We observe the limitations of ALP empirically in Table 3. Our approach, GDA, achieves more advanced scores for AGNews, IMDB, and SST-2 compared to ALP. GDA is especially effective in SST-2 showing performance increase by 19.38% for 5-shot and 13.00% for 10-shot from that of ALP.

GDA produces slot-aware rules representing shared syntactical structures across all the initial sentences sharing the same class. By applying both distance- and keyword-based manipulations, GDA generates more semantic and syntactic variations as opposed to using only one manipulation strategy. The scores of GDA(combined) in Table 3 represent the performance of GDA applied with both manipulation strategies. From this, we can observe that these rule manipulations definitively enhance the performance of GDA.

GDA achieves promising results, however, its performance is largely dependent on the existence of slot information. In other words, GDA is less effective in sentences where keywords are scarce. On the assumption that a dataset contains more keywords if it is more domain-specific, we compute the Kullback-Leibler divergence and observe that the IMDB and Yahoo datasets contain more common terms with scarce keywords than the other two datasets, i.e., the IMDB and Yahoo datasets are domain-unspecific. The major difference between the IMDB and Yahoo datasets is that almost half the sentences in the Yahoo dataset do not contain even one slot. Because GDA leverages slot information to give semantic and syntactic variation, a sparsity or lacking slots in the dataset leads to relatively poor performance of our model. We present this in Table 3, as the performance of GDA on the Yahoo dataset is relatively less sufficient in comparison to the other three datasets. We present a table of statistics and a detailed analysis of how GDA's usage of RoGs is less effective for the Yahoo dataset in Appendix D.

## 5.2 Ablation study

From the initial rules of GDA, we construct a larger set of rules with distance- and keyword-based manipulation algorithms in Section 3.2.1 and 3.2.2. We present the results of the different combinations of GDA strategies in Table 4, along with the results of the hybrid strategy utilizing both rule manipulations. However, LLMs have been regarded as effective solutions for NLP tasks, thus prompting our investigation as to whether or not GDA is more effective than LLMs for data augmentation. We employ both Bard (Thoppilan et al., 2022) and BERT (Devlin et al., 2018), widely used LLMs, to combine with GDA for augmentation and compare the performance.

|  | ATIS | | Snips | | FB-MTO | | MultiDoGO | |
| --- | --- | --- | --- | --- | --- | --- | --- | --- |
|  | 5 | 10 | 5 | 10 | 5 | 10 | 5 | 10 |
| Vanilla | 32.01 | 49.86 | 59.58 | 79.55 | 51.73 | 69.90 | 48.49 | 59.98 |
| BERT | 38.74 | 53.17 | 70.12 | **83.95** | 46.40 | 54.59 | 47.87 | 55.84 |
| Bard | 30.01 | 43.32 | 58.10 | 70.38 | 48.54 | 54.54 | 55.35 | 60.75 |
| GDA: without manipulation | 49.22 | 69.86 | 74.16 | 80.59 | 64.19 | 66.74 | 54.45 | 62.72 |
| GDA(distance): + distance manipulation | 51.83 | 69.36 | **79.25** | 75.01 | 63.01 | **70.32** | 55.28 | 62.05 |
| GDA(key): + keyword manipulation | 50.68 | 67.26 | 76.50 | 76.90 | 58.76 | 69.29 | 54.12 | 61.20 |
| GDA(combined): + both manipulation | **52.60** | **71.92** | 78.28 | 79.69 | **64.45** | 70.31 | **56.35** | **63.25** |
| GDA(combined) + BERT | 50.57 | 59.83 | 75.13 | 82.84 | 57.17 | 64.15 | 49.82 | 57.30 |
| GDA(combined) + Bard | 47.40 | 56.23 | 70.88 | 80.38 | 57.99 | 70.27 | 55.98 | 61.74 |

Table 4: Performance comparison of GDA, pre-trained language models (Bard, BERT), and their combined strategies. Each represents RoG-based, model-based, and mixed augmentation, respectively. The average of $F_1$-scores (%) is reported and bold scores are the highest scores on each setting. The score without augmentation (Vanilla) is provided for estimating the effect of augmentation strategies.

Table 4 exemplifies that the strategies of GDA are generally more effective than LLMs. For example, GDA outperforms the fine-tuned BERT by over 25% for ATIS and FB-MTO datasets. For MultiDoGO, GDA shows a 1.8% performance improvement for 5-shot and 4.1% for 10-shot compared to Bard. This proves that, the performance of GDA surpasses that of widely used LLMs. However, GDA may not always be suited for datasets such as Snips, as the results of 10-shot experiments prove less performant than that of fine-tuned BERT.

With regard to the manipulation strategies of GDA, GDA(distance) and GDA(key) represent GDA when it employs distance-based manipulation and keyword-based manipulation, respectively. GDA(combined) denotes our hybrid strategy comprised of both manipulation methods. We observe that GDA(combined) outperforms single manipulation approaches in general, an intuitive result given that GDA(combined) produces sentences of more semantic and syntactic diversity, leveraging the performance improvement of both distance- and keyword-based manipulation techniques. GDA(combined), however, does not demonstrate a significant improvement for all datasets as combining manipulation strategies does not always benefit learning performance. The performance of GDA(combined) is approximately equivalent to that of GDA(distance) for the 10-shot experiment of the FB-MTO dataset. Similarly, for the 5-shot experiment of the Snips dataset, we observe that GDA(distance) is more performant than GDA(combined). This suggests that while GDA(combined) generally performs well, its actual performance varies depending on the specific properties of a dataset.

Another experiment that was performed involves the combinations of GDA and LLMs, in which we combine both the generated sentences from GDA and from the LLMs to then train the baselines. While the combination appears promising for the 10-shot experiments of the Snips dataset, no significant improvements are observed on the rest of the datasets, as the overall performance generally declines from that of GDA(combined).

## 6 Conclusions

We have proposed a novel approach, GDA which leverages the power of RoGs and slot information and demonstrated how GDA can effectively enhance data augmentation for text classification, particularly focusing on few-shot settings. GDA solves a major issue among recent data augmentation techniques in which they potentially generate sentences with incorrect syntax or semantics, thereby limiting their effectiveness in improving performance. Our experimental results have showcased the superiority of GDA over the state-of-the-art data augmentation methods, highlighting its performance gains. Furthermore, we have conducted a qualitative analysis of the manipulation strategies inherent to GDA, specifically examining the distance- and keyword-based manipulations. Through these empirical experiments and analyses, we have confirmed that a hybrid approach combining both manipulation strategies yields the most effective results. Our approach has provided insights on the utilization of slot information and RoGs for data augmentation in text classification,

particularly within the challenging few-shot settings.

## Limitations

While GDA has demonstrated promising results and outperformed other data augmentation methods, it primarily targets few-shot settings. In one-shot learning scenarios, GDA is less effective as GDA generates sentences based on the initial RoGs. Similarly, it does not have direct applicability to zero-shot learning. Another limitation is the availability of the slot information as this information is usually limited in general datasets. We have employed language models to mitigate these limitations; however, further research into RoGs and extracting yield significant advancements.

## Acknowledgements

This research was supported by the NRF grant (RS-2023-00208094) and the AI Graduate School Program (No. 2020-0-01361) funded by the Korean government (MSIT). Han is a corresponding author. The first two authors contributed equally to this work.

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

## A    Full results of GDA

Table 5 demonstrates the performance of full strategies of GDA for the AGNews, IMDB, SST-2, and Yahoo datasets. Intuitively, named entities, used as slots, have potential noises. We confirm that the performance of the keyword-based manipulation strategy, GDA(keyword) is relatively poor compared to other rule manipulation and we can observe the result in Table 5. While the distance-based manipulation strategy, GDA(distance) shows more promising results than GDA(keyword), GDA(initial) is the most effective on average. Our results indicate that employing manipulation strategies such as GDA(keyword) and GDA(distance) for RoGs would produce more noise, resulting in lower performance.

| Dataset | AGNews | | IMDB | | SST-2 | | Yahoo | |
|---|---|---|---|---|---|---|---|---|
| $k$-shot | 5 | 10 | 5 | 10 | 5 | 10 | 5 | 10 |
| GDA(initial) | 82.66 | 82.93 | 55.21 | **76.60** | 69.95 | 75.57 | 47.19 | 52.00 |
| GDA(distance) | 82.66 | 83.96 | 60.02 | 69.48 | 70.87 | 77.18 | 47.00 | 52.50 |
| GDA(keyword) | 80.99 | 83.58 | 55.18 | 64.16 | 69.38 | 71.33 | 45.86 | 52.62 |
| GDA(keyword) + GDA(ed) | 81.03 | 83.47 | 62.57 | 69.00 | 68.12 | 73.51 | 46.94 | 52.99 |
| GDA(combined) | **83.13** | **84.22** | **67.32** | **76.60** | **75.69** | **78.78** | **49.32** | **53.06** |

5, 10: applied GDA on {5, 10}-shot train set.

Table 5: Full result. Accuracy (%) is reported.

## B    Performance comparisons of all experiment settings

Tables 6 and 7 show an overview of the experiment results we conducted. The red lines in each graph, marked by 'Full', indicate the $F_1$ scores of models trained with the full dataset,
    The augmentation settings on the scores we report are as follows:

- Vanilla: The 5- or 10-shot dataset. No augmentation is done at all.

- Initial: Vanilla + 500 sentences generated from the RoGs after the inference phase.

- Keyword, Distance, Combined: Vanilla + 500 sentences generated from RoGs on keyword-based, distance-based, and combined manipulation strategies, respectively.

- Keyword+Distance: Vanilla + 500 sentences (in total) of:
    - 250 sentences generated from RoGs on keyword-based manipulation and
    - 250 sentences generated from RoGs on distance-based manipulation.

- {Keyword,Distance,Combined}+{BERT,Bard}: Vanilla + 500 sentences (in total) of:
    - 250 sentences generated from RoGs on keyword-based (distance-based and combined, respectively) strategy and
    - 250 sentences generated from BERT or Bard.

- Full: The full, original train set. This denotes the maximum score of the model on each dataset.

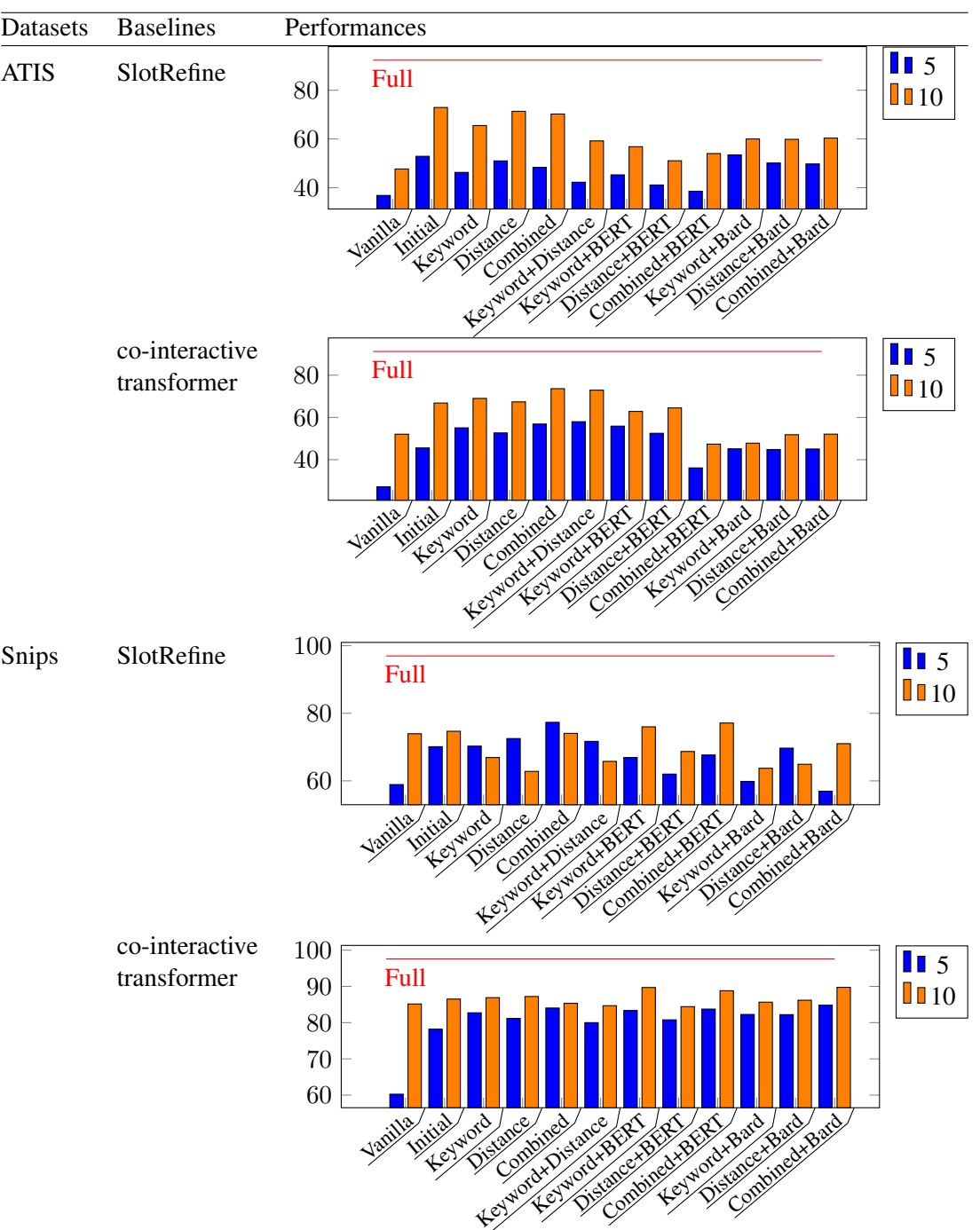

Table 6: Performance overview of the datasets with slots attached (1/2). The blue and orange bars denote the $F_1$ score (%) of each strategy on 5- and 10-shot source datasets, respectively. The red line (Full) denotes the $F_1$ score (%) of the original full dataset on each model as the expected best performance. The scores for MultiDoGO is an average of scores for six datasets.

| Datasets | Baselines | Performances |
|---|---|---|

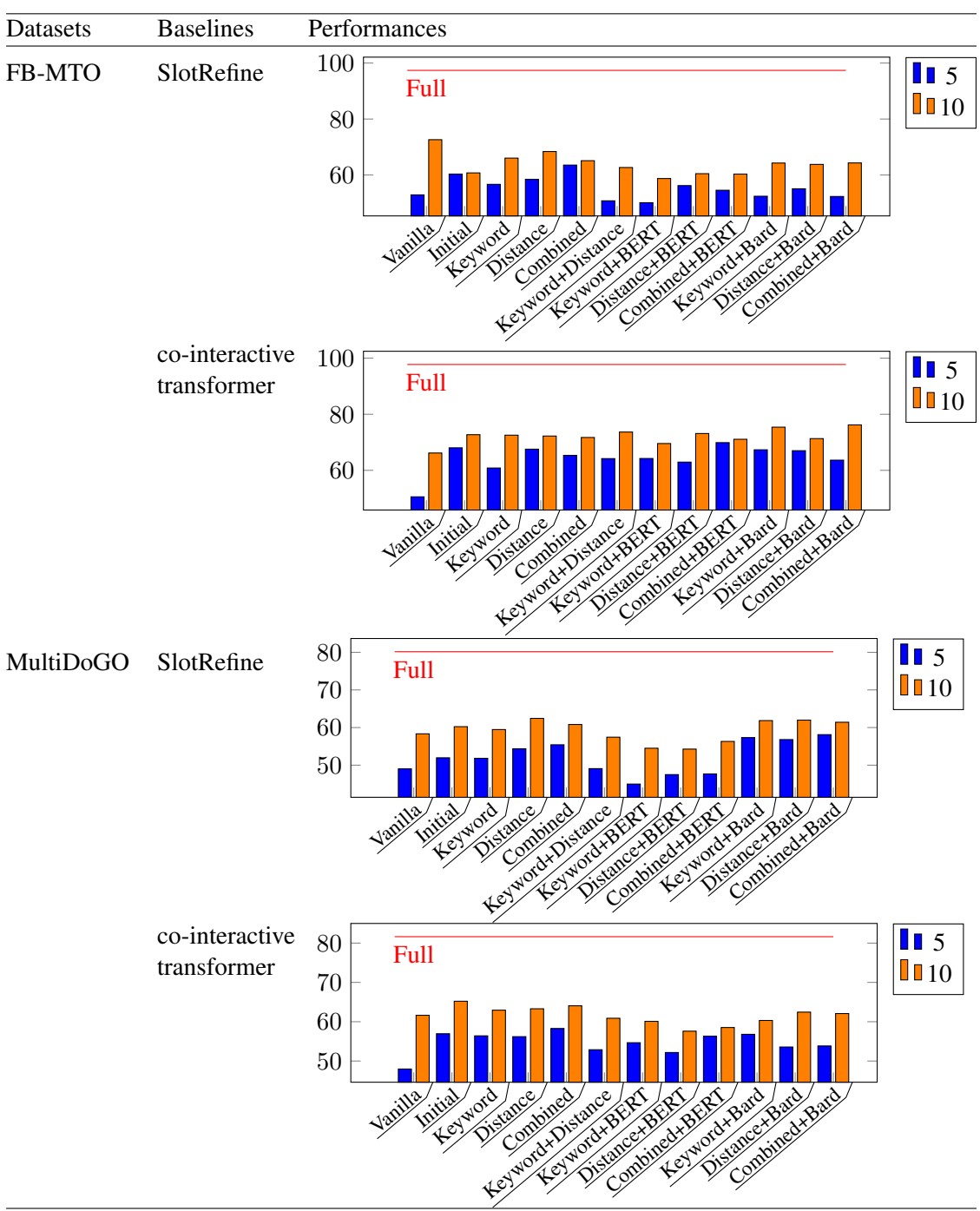

Table 7: Performance overview of the datasets with slots attached (2/2). The blue and orange bars denote the $F_1$ score (%) of each strategy on 5- and 10-shot source dataset, respectively. The red line (Full) denotes the $F_1$ score (%) of the original full dataset on each model as the expected best performance. The scores for MultiDoGO are the averages of scores for six datasets.

## C  Distance threshold on distance-based manipulations

Figure 5 illustrates the change of $F_1$ scores on each training setting by three values of a threshold Θ for normalized edit-distance. In the distance-based manipulation algorithm (Algorithm 1), DIST-CLUSTER uses this threshold value Θ to select rules whose normalized edit-distance is less than Θ and group these rules into a cluster.

Given two rules $A$ = 'add a song by $artist to $playlist' and $B$ = 'insert a song of $artist to my playlist $playlist', we compute the edit-distance by finding a token-by-token alignment that has the least number of pairs that contain different tokens. For example, we can align $A$ and $B$ as in Figure 4.

| add | a | song | by | $artist | to | [BLK] | [BLK] | $playlist |
|-----|---|------|-----|---------|-----|-------|-------|-----------|
| | | | | | | | | |
| insert | a | song | of | $artist | to | my | playlist | $playlist |

Figure 4: An example alignment of two rules.

Since the alignment in Figure 4 has the fewest number of mismatched pairs (including the pairs having [BLK] token), 4 is the edit-distance between $A$ and $B$. Then, since the normalized edit-distance is the edit-distance divided by the maximum length of either rules, $4/9 \approx 0.444$ is the normalized edit-distance between $A$ and $B$. If Θ is greater than $4/9$, these two rules will be in the same cluster.

Among the three thresholds, we observe from Figure 5 that, in the combined manipulation settings, Θ = 0.3 shows the best performance as opposed to the other threshold candidates. We presume that this value provides a good balance between interpolating the syntactic and semantic structures and introducing grammatical errors in the rules. We adopt threshold value 0.3 when we experiment on the distance-based manipulation.

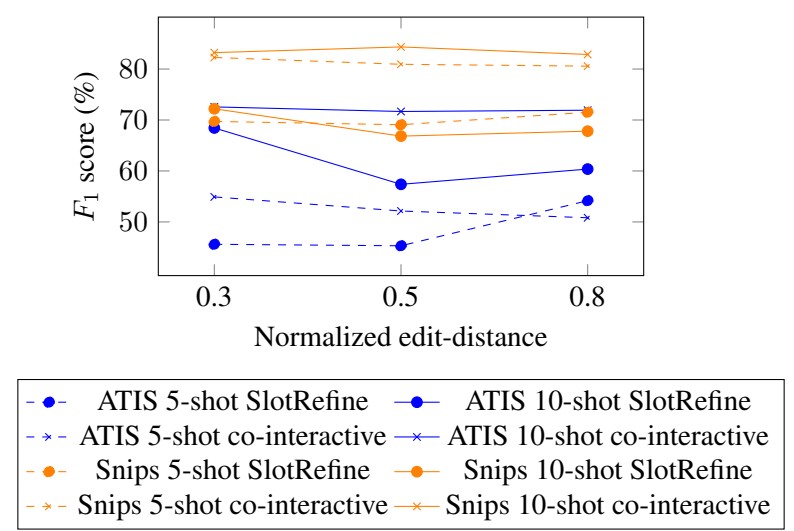

Figure 5: Performance comparison on edit-distance threshold for combined manipulation

## D  Analysis on low performance of the Yahoo dataset

The performance of GDA on the Yahoo dataset is inferior compared to the other datasets in Table 3. Among the datasets, AGNews (news), IMDB (movie review), and SST-2 (sentiment analysis) are specialized datasets for their respective purpose, but the Yahoo dataset contains general question-and-answer sentences and is not domain-specific at all. We hypothesize that domain-specific datasets would show different distributions in the usage of special terms and exploit the term frequencies of each dataset.

For two distributions $P$ and $Q$, the Kullback-Leibler divergence (KL-divergence) $D_{KL}(P \mid Q) = \sum_x P(x) \cdot \log \frac{P(x)}{Q(x)}$, the expectation of the log-likelihood from $P$ to $Q$, over $P$, estimates the difference

of the distribution $P$ from the reference distribution $Q$. The value is always non-negative, and closer to 0 when $P$ and $Q$ are similar distributions. The reference corpus we use is the British national corpus (BNC) (The BNC Consortium, 2005). BNC is one of the reference corpora for British English, which contains 100M words of written (90%) and spoken (10%) English from the 1980s to the 1990s. Since BNC would contain the typical (British) English at that time, this corpus is a reasonable reference for computing the term frequencies of general English usage.

Table 8 shows the KL-divergence value that estimates the difference from the term frequency distribution of BNC to those of datasets for each strategy.

| Dataset | $k$-shot | Augmentation strategies | | |
|---------|----------|---------|--------------|---------------|
| | | Vanilla | GDA(initial) | GDA(combined) |
| AGNews | 5 | 3.184 | 3.933 | 3.934 |
| | 10 | 2.736 | 3.303 | 3.280 |
| IMDB | 5 | 1.865 | 2.218 | 2.214 |
| | 10 | 1.404 | 1.876 | 1.832 |
| SST-2 | 5 | 3.511 | 4.165 | 4.211 |
| | 10 | 3.313 | 3.753 | 4.137 |
| Yahoo | 5 | 1.682 | 2.178 | 2.115 |
| | 10 | 1.579 | 1.957 | 2.023 |

Table 8: KL-divergence of term frequency distribution from BNC (The BNC Consortium, 2005) to each training dataset. We consider all sets with the same combination of six MultiDoGO datasets at once. 'Vanilla' column shows the performance of $k$-shot dataset before the augmentation. 'GDA(initial)' and 'GDA(combined)' columns show the performance of models trained by the sentences generated from the RoGs after the inference phase and after combined manipulation, respectively.

Table 8 shows that the AGNews and SST-2 datasets have more dissimilar term frequency distribution from the reference corpus than the IMDB and Yahoo datasets. In other words, the first two datasets are more specialized in their respective topics than the others. The following examples show augmentation results on the IMDB and Yahoo datasets.

| | |
|------|------------------------------------------------------------------|
| IMDB | I saw this film at SXSW with the director in attendance. … I've seen at this festival Frownland is among the worst. … Given that Frownland is a Captain Beefheart song maybe you'd have to be able to enjoy Trout Mask Replica on heavy rotation to appreciate this film. … |
| Yahoo | looking for fixer u property in Togo that will be sold auctioned or under HUD for extremely cheap. Gleen Falls |
| | Here's the situation My dad is currently in the hospital looking at losing his leg. … He is diabetic I don't know if that is what caused this The question is is there anything else that can be done (No slots in this text.) |

Table 9: Samples of augmentation results on the IMDB and Yahoo datasets. Applied the combined strategy on the 10-shot dataset. Some punctuations are removed after normalizing the dataset. Spacing is manually adjusted for easier reading.

The BERT NER model (Manning et al., 2014) we used, as mentioned in Section 4.1, recognizes three types of keywords: names of people (PER), places and locations (LOC), and other proper nouns (MISC). Considering that the purpose of the IMDB dataset is to predict the evaluation from its review, the text often contains several such keywords, for example, the name of directors, actors, and characters in the movie. Since GDA replaces these keywords with other words or word phrases having the same contextual information, The augmented sentences are likely to be distinct from the original sentence. On the other hand, the Yahoo dataset rarely contains keywords that NER easily recognizes (See Table 9). As a result, the Yahoo dataset has 2060 sentences out of 5100 (10 classes, 10 original sentences + 500 augmented for each class) which have no slot labels at all. Note that all 1020 sentences in the IMDB dataset have at least

one slot. This results in overfitting of the baseline models and inferior performance of the Yahoo dataset compared to other text classification datasets.