# OpenReview forum: "GDA: Grammar-based Data Augmentation for Text Classification using Slot Information"
_EMNLP/2023/Conference — EMNLP 2023 Findings_

### Official Review · Reviewer_YPDx · 2023-08-04

**Soundness:** 3

**Excitement:**

3: Ambivalent: It has merits (e.g., it reports state-of-the-art results, the idea is nice), but there are key weaknesses (e.g., it describes incremental work), and it can significantly benefit from another round of revision. However, I won't object to accepting it if my co-reviewers champion it.

**Missing References:**

None.


**Paper Topic And Main Contributions:**

The authors propose a new data augmentation method based in automatically generated generation patterns built around slot variables. This proposal is named Grammar-based Data Augmentation (GDA) and consists in 3 phases:

(1) Inference of "Rules of Grammar" (RoGs) from the training dataset. Ideally, the training dataset should have the slots already identified but, if not, a Named Entity Recognition (NER) process could be applied so the entities identified could be converted into slot variables, thus producing a slot-based dataset as needed. Taking these texts as input, basic ROGs are extracted, mere literal patterns of the examples found involving the slots.

(2) Manipulation (I would say "Combination") of the RoGs. Those patterns are combined into more general classes and the patterns themselves are combined/mixed/joined using classical lexical operations like optionality and alternation. This way, we obtain generation rules with a higher degree of variability. Three techniques are proposed for this: the use of word-level edit distance; using the slot variables as keywords to identify and combine rule groups; a mix of both.

(3) Generation. Using the rules obtained in the previous step.

The authors claim that their approach enrich the diversity of generated sentences, both at syntactic and semantic level.

Their approach is evaluated against state-of-art data augmentation methods and LLMs, also including an ablation study.



**Questions For The Authors:**

* I have problems with the use of terms such as "grammar-based", "Rule of Grammar", etc. in this work. I see no [formal] grammar here at all, what I see are just patterns, lexical patterns of some complexity that indirectly account for some syntatic and semantic variation, but lexical patterns after all. Like some kind of word-level RegExps.

* I'm concerned about the possible domain-dependendence of the approach. It reminds me to some classic  Information Extraction (IE) algorithms for automatically generating extraction patterns through bootstrapping.

* Why are doing the ablation analysis (Table 4) on different datasets that the otiginal ones used for performance comparison (Table 3). The logical action would be to use the same datasets for both studies.


**Reasons To Accept:**

* Data augmentation is a hot topic. It's difficult to find or create datasets good and large enough for current data-based NLP approaches.


**Reasons To Reject:**

* I see no breakthrough of possible interest for the NLP community.


**Reproducibility:**

4: Could mostly reproduce the results, but there may be some variation because of sample variance or minor variations in their interpretation of the protocol or method.

**Reviewer Confidence:**

3: Pretty sure, but there's a chance I missed something. Although I have a good feel for this area in general, I did not carefully check the paper's details, e.g., the math, experimental design, or novelty.

**Typos Grammar Style And Presentation Improvements:**

* Abstract is excesively long, it must be reduced.
* Sect.1: this section  should be re-structured, probably split in two. Basically, all content in the right column of page 2 (including Figure 2) is redundant or goes beyond the contents we expect from this section.
* Sect. 1: I also miss a short paragraph describing the structure of the paper.

* Sect. 3.1: I see no reason for explaining CFGs, NLP community knows them very well.

* Sect. 4.1: A brief description of the nature and domain of the documents of those datasets may be of interest to the reders. In my case, surely it is.

* Table 3: I suggest saying in the caption that those results correspond to the datasets WITHOUT slots. The contrary for Table 4.

* Sect. 5.1, par. 1: It is not clear that "GDA achieves compeling performance compared to baseline augmentation models". I see no clear superiority with respect to ALP, for example.

* References: I suggest not listing the whole list of authors in the case of the entries (Brown et al., 2020) and (Thoppilan et al 2022). Just list part of them foillowed by "et al."

---

> ### Author Rebuttal · Authors · 2023-08-29
>
> We are appreciative of the reviewer's extensive remarks and proposals. We value the beneficial criticism that will upgrade our paper. We shall ensure that your input is incorporated in the revised version.
>
> **1. Contribution of GDA**
>
> >Reasons to Reject) I see no breakthrough of possible interest for the NLP community.
>
> **[Response]** In a few sentences, GDA constructs rules and by the proposed rule manipulations, generates sentences which enrich the syntactic and semantic diversity - contributing to data augmentation and a few-shot text classification research field. Furthermore, our grammar-based approach acts as a helpful tool for neuro-symbolic approaches, enhancing the interpretability of the result to deep learning models. We find our results interesting and well-performing enough to be an impactful contribution in the NLP community. We would appreciate more detailed explanations of why the reviewer thinks our paper has no significant breakthroughs.
>
> **2. Clarification on the formal grammar used in the paper**
>
> >Question #1) I have problems with the use of terms such as "grammar-based", "Rule of Grammar", etc. in this work. I see no [formal] grammar here at all, what I see are just patterns, lexical patterns of some complexity that indirectly account for some syntatic and semantic variation, but lexical patterns after all. Like some kind of word-level RegExps.
>
> **[Response]** The formal grammar used in the paper is context-free grammar (CFG). Our ruleset of GDA is based on CFG. The distance-based rule manipulation is also based on the edit-distance metric, a popularly used similarity measure in the theory of computation. Roughly speaking, every formal grammar is based on patterns. So if the reviewer disagrees, I respectfully ask the reviewer what their definition of formal grammar is if CFG does not fall into this category. We would be happy to provide additional information on the definition of formal grammar.
>
> **3. Dependence on slot information and how we stated that our approach also gains improvements on non-domain-specific general dataset**
>
> >Question #2) I’m concerned about the possible domain-dependendence of the approach. It reminds me to some classic Information Extraction (IE) algorithms for automatically generating extraction patterns through bootstrapping.
>
> **[Response]** We acknowledged our own concerns regarding GDA’s dependence on slot information. As a result, we not only conduct experiments on datasets with slot information but also on datasets without slot information. Table 3 presents the performance result of GDA on datasets without designated slot information. We use a pre-trained BERT for NER task to extract named entities and use them in substitution of slot information. The result in Table 3 shows that GDA gains objective improvements on general datasets.
> Furthermore, our rules and manipulations require sophisticated computational steps depicted in Algorithms 1-3, which are not possible solely with just IE algorithms. If the reviewer finds it lacking, we would like to know the detailed comments on this concern. We would be happy to respond to your feedback.
>
> **4. Clarification on how the purpose of the two experiments is different**
>
> >Question #3) Why are doing the ablation analysis (Table 4) on different datasets that the otiginal ones used for performance comparison (Table 3). The logical action would be to use the same datasets for both studies.
>
> **[Response]** Our experiments consist of 1) comparison of baseline data augmentation (DA) models for general datasets without slot information and 2) comparison results of language models and various combinations in GDA for datasets with slot information.
> Our approach achieves high performance for experiment 2, but we are also concerned as to whether our approach is also applicable for general text classification datasets with no designated slot information.
> The purpose of the experiment 1 is to inspect whether GDA can achieve state-of-the-art performance even on the general datasets for text classification that do not contain designated slot information.Table 3 shows the performance improvements compared to the baseline DA models.
>
> On the other hand, experiment 2 primarily focuses on how GDA, the rule-based data augmentation method, performs better than the language models. We also inspect various combinations of GDA from this set of datasets. The focus of experiment 2 is more centered on the hybrid strategies of GDA in combination with the language models.
> We will make sure to clearly describe the different intentions of both experiments in the final version of our paper.
>
>
> We highly appreciate your valuable feedback and insightful perspectives, which offer new angles to consider in relation to the detailed comments on GDA. We are confident that our response will successfully alleviate any concerns you might have about the robustness of our approach. Should you find our explanations lacking, please let us know. We are eager to provide you with the essential information needed to comprehensively address your inquiries.

---

### Official Review · Reviewer_VFyi · 2023-08-07

**Soundness:** 3

**Excitement:**

3: Ambivalent: It has merits (e.g., it reports state-of-the-art results, the idea is nice), but there are key weaknesses (e.g., it describes incremental work), and it can significantly benefit from another round of revision. However, I won't object to accepting it if my co-reviewers champion it.

**Paper Topic And Main Contributions:**

This paper proposes a new approach to data augmentation, designed specifically to take advantage of slot information, where that exists. The approach is called "Grammar-based Data Augmentation", where "grammar" refers to the use of a generative, CFG-style set of production rules to - in essence - transform original input sentences and produce new, similar sentences with similar semantic content to the originals. The process of generating data for augmentation involves three steps: 1) collecting a set of production rules from the slot-tagged input data (the authors refer to this as "inference", but from my perspective they are harvesting rules, not inferring them); 2) building up a set of substitution rules/templates, by perform various manipulations on the production rules collected in step #1; and 3) generation of new sentences according to the templates produced in step #2.

The paper presents two sets of experiments, with different evaluation metrics and comparisons. Both sets of experiments include ablations comparing GDA with and without the two different types of manipulations described as part of step #2:

Set A (results in Table 3): application of GDA to four datasets without original slot labeling; NER is used as a stand-in for slot information (as the steps described above crucially rely on having slot information). In these experiments, GDA is compared to three SOA data augmentation strategies. Performance is reported in accuracy of the relevant text classification task.

Set B (results in Table 4): application of GDA to four datasets with original slot labeling. In these experiments, GDA is compared to data augmentation using pretrained LLMs (both BERT and Bard), but is not compared to the three SOA data augmentation strategies from the Set A experiments. Performance is reported in F1 for the relevant text classification task.

Overall, some of the results presented here do seem to achieve better performance than the state-of-the-art, and the method proposed can easily be applied to any dataset with existing slot information, with rather low computational overhead. That said, it is difficult to draw convincing conclusions, given the differences in experimental settings, which are not, in my opinion, convincingly motivated. See detailed points below.

**Questions For The Authors:**

A. line 200/201 and after: the use of the word "variables" here is, in my experience, unusual. I would suggest instead talking about "terminals" and "non-terminals", in keeping with traditional CFG terminology.

B. lines 255-257: in what way does this approach ensure that underlying semantics and grammatical structure are preserved?

C. section 3.2.1 is confusing to me: the edits proposed seem to be okay, but I find this description confusing and somewhat misleading. it's not easy to see why these should be called "distance-based", nor what edit-distance has to do with producing these types of rules. Is the idea that you're taking two sequences/sentences that have a similar ordering of slots (and perhaps the same intent?) and generating a rule that could produce either sequence, using traditional edit operations? This part of the approach is not as clearly explained as most other parts.

D. section 3.2.2: what is the definition of "keyword" being used here? It's a big claim to say that "keywords imply the main intention of a sentence", especially without explaining what counts as a keyword in general. (I understand that slots are taken to be keywords under this approach, but that is an approximation - how would you define "keyword" more generally?)

E. Algorithm 2: do the rules grouped here need to share the same class?

F. line 465: over what are you computing KL divergence? More detail would be helpful here.

G. line 555: the conclusion mentions qualitative analysis, but I don't see any qualitative analysis in the body of the paper. I'm sure I missed something - to what does this refer?

Finally, thanks for your good work!

**Reasons To Accept:**

The main strength of this paper is the development of a CFG-based approach to building sentence generation templates for performing data augmentation for text classification, particularly for datasets which containing slot information. This approach has low computational requirements and seems to compare favorably to other approaches (though this result is not adequately demonstrated).

I very much like the motivation of this approach, and I believe it has promise to be an effective and accessible data augmentation method, which could be demonstrated through additional experiments and analyses.

In addition, the paper is well-organized and mostly well-written.

**Reasons To Reject:**

I see three main weaknesses to this paper - these are described in more detail below. In my opinion, the weaknesses are strong enough to outweigh the strengths described above.

1. Inconsistency of experimental settings and reporting of experimental results.

The paper claims that GDA, which is designed specifically to use slot information encoded in existing datasets, outperforms other similar data augmentation approaches (specifically, EDA, SSMBA, and ALP), while avoiding "semantic errors" and inappropriate changes to sentence structure. The experiments presented, though, only compare to these systems on datasets without original slot information, and for which slot information has been approximated using automated named entity labeling. There is a second set of experiments over the claimed target data type - datasets with slot information labeled - but these experiments do *not* compare to the other DA methods described in the paper. Instead, they compare to use of LLMs for generating new data. There is no explanation in the paper of why the second set of experiments does not include comparisons to EDA, SSMBA, and ALP, and I cannot myself come up with a reasonable justification for this omission.

In addition, the results that do show improvements over comparable SOA methods (those reported in Table 3) report performance with respect to accuracy rather than F1, again without a clear motivation. These factors combined reduce my confidence in the results reported, because they are not motivated. A more minor point - I would question some of the claims made about GDA outperforming other systems, as no significance testing seems to have been done. For example, the paper claims that GDA outperforms ALP for the IMDB dataset, but the difference is tiny - 67.32 accuracy compared to 67.05.

2. Overly strong argumentation regarding elimination/reduction of semantic errors.

Starting from the abstract, the paper claims that GDA, by incorporating grammar rules, reduces the occurrence of certain semantic errors common to comparable DA approaches. The only support for this claim is through the examples shown in Table 1. Even with these carefully selected examples, the errors shown are not all semantic errors, and generally are not due to changes in sentence structure. Instead, they're because of either ill-formed NPs ('Bruno Hits Mars') or bad selectional restrictions ('add my playlist to a song') -- even in the latter case, the structure of the sentence doesn't change. The Table 1 errors are claimed to be logical errors, but they are not at all consistent error types. The two ALP examples are semantically fine, in my opinion. SSMBA has one logical error (can't add a playlist to a song) and one PP-attachment error. EDA has one factual error (not equivalent to a logical error) and one clunky (but possible) NP structure.

On another point, the abstract claims that the "rules of grammar" produced by the GDA approach represent "relations between a sentence's semantics and its syntactical structure". I can only read this as true with a very loose definition of what is meant by both "semantics" and "syntactical structure" -- as a linguist, this wording leads me to expect a representation that actually incorporates syntactic constraints (or at least syntactic classes) and some sort of semantic representation. The only semantic representation I see, though, is the use of (rather domain-specific) informational categories as slot labels (e.g. $artist or $playlist). The rules produced look like sophisticated templates for generating text; they do not represent either syntactic or semantic information in the usual sense (again, I'm speaking from the perspective of linguistics).

3. Some claims not supported by results.

a. Claim about types of errors made by GDA vs. those made by other comparable approaches. The paper claims several times that GDA reduces semantic errors, but does not demonstrate that. Around lines 448-457, the paper claims that GDA generates more semantic and syntactic variations, but this is not demonstrated, at least not that I see. It would be interesting to perform a study of the sets of sentences generated by the various approaches based on the same input data!

b. Claim that Table 3 demonstrates improvements of GDA over ALP. This feels like a stretch - performance of GDA is better for some data sets, but only some (mostly just SST-2, where evaluation is happening on a fairly small set. Also, not clear how this strategy really differs from ALP. If ALP also uses CFGs, this should have been noted much earlier - the paper leans toward presenting use of CFG-like structures as something novel.

c. No significance testing - some of the differences claimed to show better performance, whether compared to another model or comparing across different datasets, seem to be very very small.

**Reproducibility:**

3: Could reproduce the results with some difficulty. The settings of parameters are underspecified or subjectively determined; the training/evaluation data are not widely available.

**Reviewer Confidence:**

3: Pretty sure, but there's a chance I missed something. Although I have a good feel for this area in general, I did not carefully check the paper's details, e.g., the math, experimental design, or novelty.

**Typos Grammar Style And Presentation Improvements:**

* abstract: spell out full form of 'GDA'

* line 175 (compared to abstract): the abstract talks about using "context-aware grammar", but the rest of the paper talks about using "context-free grammar". The rules and approach described are certainly context-free - is there a reason for calling this approach "context-aware" in the abstract? (or maybe just a typo or missed edit...)

* section 4.1: the paper would benefit from more discussion of the datasets used, and in particular what their respective tasks are, and what the set of possible classes looks like for each. the different DA approaches discussed in the paper use different strategies (for example, changes in sentence structure vs. changes in the structure of NPs vs. choice of verb), and different strategies may be more or less useful for different tasks.

* Section 5: I don't understand why there is so little discussion of the results in Table 4, and so much discussion of results in Table 3. The main contribution of the paper is a DA method that relies on slot information, and yet the bulk of the discussion of results focuses on the experiments on datasets that do not include slot information. This is confusing.

---

> ### Author Rebuttal · Authors · 2023-08-29
>
> We would like to begin by expressing our appreciation for the thorough comments and proposals. We are thankful for the productive remarks that could refine our paper. We assure you that we will incorporate your feedback in the final version.
>
> **1. Details on different types of experiments in the paper**
>
> >Reasons to Reject #1) Inconsistency of experimental settings and reporting of experimental results.
> The paper claims that GDA, which is designed specifically to use slot information encoded in existing datasets, outperforms other similar data augmentation approaches (specifically, EDA, SSMBA, and ALP), while avoiding "semantic errors" and inappropriate changes to sentence structure. The experiments presented, though, only compare to these systems on datasets without original slot information, and for which slot information has been approximated using automated named entity labeling. There is a second set of experiments over the claimed target data type - datasets with slot information labeled - but these experiments do not compare to the other DA methods described in the paper. Instead, they compare to use of LLMs for generating new data. There is no explanation in the paper of why the second set of experiments does not include comparisons to EDA, SSMBA, and ALP, and I cannot myself come up with a reasonable justification for this omission.
> In addition, the results that do show improvements over comparable SOA methods (those reported in Table 3) report performance with respect to accuracy rather than F1, again without a clear motivation. These factors combined reduce my confidence in the results reported, because they are not motivated. A more minor point - I would question some of the claims made about GDA outperforming other systems, as no significance testing seems to have been done. For example, the paper claims that GDA outperforms ALP for the IMDB dataset, but the difference is tiny - 67.32 accuracy compared to 67.05.
>
> **[Response]** There are two sets of datasets: those without slot information and those with slot information. Our approach achieves high performance for the datasets with slot information, but we are also concerned whether our approach is also applicable for general text classification datasets with no designated slot information.
> The purpose of the experiment of the datasets without slot information is to inspect whether GDA can achieve state-of-the-art performance even on the general datasets for text classification that do not contain designated slot information. Table 3 highlights the performance improvements compared to the baseline DA models.
> On the other hand, experiments on the datasets including slot information primarily focus on how GDA, the rule-based data augmentation method, performs better than the language models. We also inspect various combinations of GDA from this set of datasets. The focus of experiments for the datasets with slot information is more centered on the hybrid strategies of GDA in combination with the language models.
> We did not think it fair to compare the baseline DAs to GDA because they do not make use of slot information. However, we will make sure to clearly describe the different intentions of the experiments for the sets of datasets including and excluding slot information in the final version.
>
> **2. Explanation of strengths of GDA in syntactic and semantic diversity & error correction in a linguistic manner**
>
> >Reasons to Reject #2) Overly strong argumentation regarding elimination/reduction of semantic errors.
>
>
> Starting from the abstract, the paper claims that GDA, by incorporating grammar rules, reduces the occurrence of certain semantic errors common to comparable DA approaches. The only support for this claim is through the examples shown in Table 1. Even with these carefully selected examples, the errors shown are not all semantic errors, and generally are not due to changes in sentence structure. Instead, they're because of either ill-formed NPs ('Bruno Hits Mars') or bad selectional restrictions ('add my playlist to a song') -- even in the latter case, the structure of the sentence doesn't change. The Table 1 errors are claimed to be logical errors, but they are not at all consistent error types. The two ALP examples are semantically fine, in my opinion. SSMBA has one logical error (can't add a playlist to a song) and one PP-attachment error. EDA has one factual error (not equivalent to a logical error) and one clunky (but possible) NP structure.
>
>
> On another point, the abstract claims that the "rules of grammar" produced by the GDA approach represent "relations between a sentence's semantics and its syntactical structure". I can only read this as true with a very loose definition of what is meant by both "semantics" and "syntactical structure" -- as a linguist, this wording leads me to expect a representation that actually incorporates syntactic constraints (or at least syntactic classes) and some sort of semantic representation. The only semantic representation I see, though, is the use of (rather domain-specific) informational categories as slot labels (e.g. artist | playlist). The rules produced look like sophisticated templates for generating text; they do not represent either syntactic or semantic information in the usual sense (again, I'm speaking from the perspective of linguistics).
>
> >Reasons to Reject #3-a,b) Some claims not supported by results.
> a. Claim about types of errors made by GDA vs. those made by other comparable approaches. The paper claims several times that GDA reduces semantic errors, but does not demonstrate that. Around lines 448-457, the paper claims that GDA generates more semantic and syntactic variations, but this is not demonstrated, at least not that I see. It would be interesting to perform a study of the sets of sentences generated by the various approaches based on the same input data!
> b. Claim that Table 3 demonstrates improvements of GDA over ALP. This feels like a stretch - performance of GDA is better for some data sets, but only some (mostly just SST-2, where evaluation is happening on a fairly small set. Also, not clear how this strategy really differs from ALP. If ALP also uses CFGs, this should have been noted much earlier - the paper leans toward presenting use of CFG-like structures as something novel.
>
> **[Response]** While it’s true that the sentences in Table 1 do not completely exemplify grammatical or semantic error, such logical errors still critically downgrade the performance of baselines and are thus harmful to the classification performance. The reviewer pointed out that the sentence examples of ALP are not sufficient for showing semantic errors of ALP. Given the following 5 sentences, I’ll provide the sentences that ALP and GDA generate:
>
> * i'd like flight information from dallas fort worth to boston on tuesday,
>
> * list all afternoon flights on united airlines from san francisco to denver,
>
> * is there an atlanta flight to denver connecting,
>
> * i want to fly boston to dallas,
>
> * tell me the last flight from atlanta to philadelphia.
>
> The following examples are the sentences that ALP generates:
>
> * Dallas Fort Worth to Boston flight information on Tuesday,
>
> * List all afternoon flights on United Airlines Dallas Fort Worth to Boston,
>
> * Is there a direct flight from Atlanta to Denver,
>
> * Fly Boston to Dallas,
>
> * Atlanta flight to Denver connecting.
>
> The above sentences might look contextually right in the perspective of humans, but the grammatical errors in sentences such as “Fly Boston to Dallas” and “Atlanta flight to Denver connecting” are unlikely to be golden data.
>
> The following examples are the sentences that GDA generates:
>
> * list all afternoon flights on united airlines from atlanta to philadelphia,
>
> * I'd like flight information from atlanta to denver connecting,
>
> * Tell me the flight from boston to dallas,
>
> * Is there flight information from dallas fort worth to boston,
>
> * I want to fly san francisco to denver on tuesday.
>
> While the above sentences maintain the grammatical correctness, they enrich the syntactic representation, contributing to the strong performance of GDA.
>
> While we make the claim that each baseline has its own corresponding weakness, we also state the semantic and syntactic diversity that GDA provides. We are grateful that the reviewer understands the core of our initial rule generation. We depend on the slot information (either represented with slot labels or named entities) for rule generation. The initial ruleset, as the reviewer pointed out, lacks the representation of semantic and syntactic information. Thus, we use rule manipulations to enhance the weakness of the initial ruleset.
> From rule manipulations, we first find similar sentences both in their structures and semantics. Then, by distance-based manipulation, we align words based on the edit-distance metric and group the sequences of words that can be used to replace one another. Example rules for distance-based manipulation are shown in Figure 3(a). When utilizing keyword-based manipulation, we group the sequences based on the slot information. This example rule for keyword-based manipulation is shown in Figure 3(b). The rules show the syntactic information of possible variants that GDA can generate. We believe the syntactic information of GDA rules guarantees the corresponding syntactic diversity that enhances classification performance. However, as we are approaching this from a formal language and computer science background we would be grateful if the reviewer would comment about a more sophisticated representation of syntactic information and the connotation of syntactic diversity from a linguistic perspective.
>
> We find slot information is a good representation of the semantic information that rules carry.
> If the reviewer disagrees, it would be helpful and we would appreciate it if the reviewer would provide which representation would be adequate for semantics.
>
>
> **3. Clarification on the interpretation of the result**
>
> >Reasons to Reject #3-a,c) Some claims not supported by results.
> a. Claim about types of errors made by GDA vs. those made by other comparable approaches. The paper claims several times that GDA reduces semantic errors, but does not demonstrate that. Around lines 448-457, the paper claims that GDA generates more semantic and syntactic variations, but this is not demonstrated, at least not that I see. It would be interesting to perform a study of the sets of sentences generated by the various approaches based on the same input data!
> c. No significance testing - some of the differences claimed to show better performance, whether compared to another model or comparing across different datasets, seem to be very very small.
> >Question F) line 465: over what are you computing KL divergence? More detail would be helpful here.
>
> **[Response]** We use Tables 3 and 4 to highlight and empirically confirm that GDA is effective, and also present sentences generated by the baselines and GDA in Table 1 for comparison. We agree that case studies on sentence generation would be helpful to intuitively understand the result of GDA, however due to the space limit, we did not present such a study regarding the comparison of actual data. We appreciate your suggestion and will make sure to add an additional page presenting a case study on comparing sentences generated by the baselines and GDA in the final version of our paper.
>
> We appreciate the reviewer’s comment on the performance in Table 3. ALP performs better for the 10-shot experiment on AGNews and experiments of Yahoo while GDA performs better for IMDB, SST-2, and 5-shot of AGNews. GDA performs especially well for SST-2, however,2, however, the performance of GDA for Yahoo shows a large performance gap even compared to other baselines. The reason for this is that sentences in the Yahoo dataset scarcely contain named entities. Through analysis of KL divergence on the distribution of total words and slot information (named entities for the Yahoo dataset) from the Yahoo dataset, we observed that the sentences from the dataset scarcely contain words with named entities. We present a detailed case analysis of KL divergence for the Yahoo dataset in Appendix D.
>
>
> **4. Detailed description of our rule inference and manipulation**
>
> >Question B) lines 255-257: in what way does this approach ensure that underlying semantics and grammatical structure are preserved?
>
> >Question C) section 3.2.1 is confusing to me: the edits proposed seem to be okay, but I find this description confusing and somewhat misleading. it's not easy to see why these should be called "distance-based", nor what edit-distance has to do with producing these types of rules. Is the idea that you're taking two sequences/sentences that have a similar ordering of slots (and perhaps the same intent?) and generating a rule that could produce either sequence, using traditional edit operations? This part of the approach is not as clearly explained as most other parts.
>
> >Question D) section 3.2.2: what is the definition of "keyword" being used here? It's a big claim to say that "keywords imply the main intention of a sentence", especially without explaining what counts as a keyword in general. (I understand that slots are taken to be keywords under this approach, but that is an approximation - how would you define "keyword" more generally?)
>
> >Question E) Algorithm 2: do the rules grouped here need to share the same class?
>
> >Question G) line 555: the conclusion mentions qualitative analysis, but I don't see any qualitative analysis in the body of the paper. I'm sure I missed something - to what does this refer?
>
> **[Response]** The key distinction between ALP and GDA is rooted in the usage of slot information and rule manipulation.  Our approach may look similar to ALP, as our ruleset is based on context-free grammar (CFG), however diverging from the basic form of the rules, our main focus is the slot information and rule manipulation.
> We use slot information (e.g., slot labels and named entities) to group the words with the same type of information. For instance from Figure 2, \\$artist represents identifying slot information for the artists (e.g., Adele, Bruno Mars, IU). The initial ruleset consists of 1) rules that map slot information to the corresponding words and 2) rules for the structure of each sentence - words replaced with slot information (or named entities).
>
> While the initial ruleset is not unlike the initial sentences - save the slot information representation, we use distance- and keyword-based manipulation to enrich the ruleset. The distance-based rule manipulation is based on the edit-distance metric. We take into account the length of each sentence in order to normalize the edit-distance metric (normalized ED) and calculate the similarity between sentences. The similarity between sentences is calculated by using normalized ED and then by the predefined threshold, we can group the similar rules as exemplified in Algorithm 1. Algorithm 1 details how we calculate normalized ED and align the grouped sentences to merge the rules. For instance, in Figure 3 when we apply distance-based manipulation for the four initial rules, we first align sequences of words that minimize normalized ED and merge the four rules to make a new superset-like rule. The resulting rule is in Figure 3(a).
>
> Keyword-based rule manipulation is less complex. We gather the sentences with the same type and order of slot information. We then use slot information which are present in the form of slot labels for datasets such as ATIS, Snips, FB-MTO, and MultiDoGO, and named entities for datasets such as AGNews, IMDB, SST-2, and Yahoo. For instance in Figure 3, the four initial rules have \\$artist and \\$playlist as slot information, and \\$artist occurs first and then \\$playlist occurs. Thus, with the same type and order of slot information, we group the four initial rules by aligning the slot information. The resulting rule can be seen in Figure 3(b).
>
> Section 5.2 provides ablation studies and detailed analyses on manipulations of GDA and hybrid strategies of GDA and language models. Providing the full experimental results in Appendix B, we supply more details of the analyses in Appendices C and D. If the reviewer finds our analyses insufficient, we would appreciate getting feedback so that we might answer your concerns.
>
> **5. Clarifying general issues**
>
> &nbsp; **a.** The terminology of CFG
>
> >Question A) line 200/201 and after: the use of the word "variables" here is, in my experience, unusual. I would suggest instead talking about "terminals" and "non-terminals", in keeping with traditional CFG terminology.
>
> **[Response]** We follow the CFG terminology as utilized in theory of computation [1], which uses 'variable'. However, we are grateful of the reviewer’s detailed concerns of the terms.
>
> [1]: Michael Sipser, "Introduction to the theory of computation", 2012.
>
> &nbsp; **b.** The grammar and rules of GDA
>
> >Comments #1) Line 175 (compared to abstract): the abstract talks about using "context-aware grammar", but the rest of the paper talks about using "context-free grammar". The rules and approach described are certainly context-free - is there a reason for calling this approach "context-aware" in the abstract? (or maybe just a typo or missed edit...)
>
> **[Response]** We use the term, ‘context-aware grammar’, to indicate that the rules of GDA extend from context-free grammar to represent the context of sentences, thus making them ‘context-aware’. While the basic format of a ruleset generated by GDA is CFG, we design the rules of GDA to be aware of context. The usage of slot information in rule manipulation maintains the abstractive context of rules and restricts the generation of rule variants that may harm the original intention of the rules. We will make sure to be clear in describing the ruleset of GDA, and try to refrain from using unclear terminology in order to avoid any confusion.
>
> &nbsp; **c.** Discussion of the results
> >Comments #2) Section 5: I don't understand why there is so little discussion of the results in Table 4, and so much discussion of results in Table 3. The main contribution of the paper is a DA method that relies on slot information, and yet the bulk of the discussion of results focuses on the experiments on datasets that do not include slot information. This is confusing.
>
> **[Response]** We provide comparison results with DA baselines for general datasets without slot information (1) and comparison results of language models and various combinations in GDA for datasets with slot information (2).
>
> Table 3 shows the performance of (1) and we present an analysis of the experiments in Section 5.1. Table 4 shows the performance of (2) and we present an analysis of the experimental results in Section 5.2. We aim to provide the intentions of the analyses and provide detailed descriptions for comprehensible interpretation of results for both experiments. Thus, we strive to present a similar amount of analysis of experiments for both (1) and (2) in the paper. We would appreciate it if the reviewer would provide details of what the analysis of (2) is lacking. If we need more explanation of Table 4 for the readers to understand, we will use an additional page for the analysis in the final version.
>
> We value your insights and expertise, which encompass novel perspectives on the comprehensive feedback concerning GDA. Our rebuttal aims to address any reservations about the validity of our approach. Should you find our explanations insufficient, please feel free to inform us. We are enthusiastic about providing the necessary information to address your queries effectively.

---

### Official Review · Reviewer_2Dha · 2023-08-10

**Soundness:** 3

**Ethical Concerns:**

Yes

**Excitement:**

4: Strong: This paper deepens the understanding of some phenomenon or lowers the barriers to an existing research direction.

**Paper Topic And Main Contributions:**

This paper introduces a novel approach called "Guided Data Augmentation" (GDA) for enhancing data augmentation techniques in the context of text classification, particularly focusing on few-shot learning scenarios. The main problem addressed by the paper is the limitations of existing data augmentation methods in generating sentences with accurate syntax and semantics, which hinders their effectiveness in improving text classification performance, especially in scenarios with limited training data.

**Reasons To Accept:**

- Enhanced Data Augmentation Strategies: The paper introduces manipulation strategies that enhance both semantic and syntactic diversity in augmented data. This innovation is particularly important in preventing incorrect syntax and semantics in generated sentences, improving the overall quality of the augmentation process.

- The approach demonstrates significantly improved performance compared to other existing techniques, essentially presenting a enhanced iteration of EDA (Easy Data Augmentation).

**Reasons To Reject:**

The limitations of this paper include the dependence on slot information, making it less effective for datasets lacking such information. Additionally, the approach may not be suitable for one-shot learning scenarios and zero-shot learning, potentially limiting its applicability. There's also a risk that the presented hybrid strategy may not consistently outperform single manipulation approaches, leading to variable performance across different datasets. The reliance on named entities extracted by a language model for slot information could introduce noise or inaccuracies, affecting the overall effectiveness of the technique.

**Reproducibility:**

3: Could reproduce the results with some difficulty. The settings of parameters are underspecified or subjectively determined; the training/evaluation data are not widely available.

**Reviewer Confidence:**

3: Pretty sure, but there's a chance I missed something. Although I have a good feel for this area in general, I did not carefully check the paper's details, e.g., the math, experimental design, or novelty.

---

> ### Author Rebuttal · Authors · 2023-08-29
>
> We are grateful to the reviewer for the extensive comments and recommendations. We value constructive remarks that could enhance our paper. We will make sure to include your feedback in the revised version.
>
> **1. Dependence on slot information (e.g., named entities, slots)**
>
> >Reasons to Reject #1) The limitations of this paper include the dependence on slot information, making it less effective for datasets lacking such information. There's also a risk that the presented hybrid strategy may not consistently outperform single manipulation approaches, leading to variable performance across different datasets. The reliance on named entities extracted by a language model for slot information could introduce noise or inaccuracies, affecting the overall effectiveness of the technique.
>
> **[Response]** Slot information plays an important role in the distance- and keyword-based manipulation strategies employed by GDA when constructing rules. However, there is not a dependency on designated slot information in the data. We use a pre-trained BERT to extract named entities, which are then utilized as slot information, and show how GDA performs well with general datasets such as AGNews and IMDB in Table 3.
>
> As noted by the reviewer, it is true that the hybrid approach of using language models to extract named entities might produce unexpected noises, an issue that we illustrate in Section 7. However, while our approach is useful for both datasets with and without slot information, we are cautious of employing language models solely for the purpose of yielding slot information. Thus, as described in Section 7, we aim to incorporate rule-based algorithms to solve the potential danger of employing language models for non-slot information related tasks.
>
> **2. The limitation of GDA for zero- and one-shot learning**
>
> >Reasons to Reject #2) Additionally, the approach may not be suitable for one-shot learning scenarios and zero-shot learning, potentially limiting its applicability.
>
> **[Response]** Recent research on few-shot text classification is mainly focused on 5- and 10-shot settings as securing several sentences for each class are guaranteed in this field of research [1-3]. Our paper also aims for the few-shot text classification where several sentences for each class are given. Thus, zero- and one-shot learning are not prioritized in this paper. We do, however, acknowledge this limitation in Section 7. We cannot solely use GDA for zero- and one-shot learning, but we can employ a language model for a few initial sentences and then use GDA in some sort of hybrid strategy. We’ll issue a more detailed explanation on how to use GDA for zero- and one-shot learning in the final version.
>
> [1] Kim et al., ALP: Data Augmentation Using Lexicalized PCFGs for Few-Shot Text Classifcation, AAAI’22.
>
> [2] Sun et al., Hierarchical Attention Prototypical Networks for Few-Shot Text Classification, EMNLP’19.
>
> [3] Lyu et al., Few-Shot Text Classification with Edge-Labeling Graph Neural Network-Based Prototypical Network, Coling’20.
>
> Overall, we acknowledged the limitations of GDA that the reviewer expressed concerns about in Section 7.
> We are grateful for your valuable insights and perspectives, particularly the detailed observations on GDA. We aim to address your concerns regarding the robustness of our approach through our rebuttal. If you find our responses insufficient, please let us know, and we would be pleased to provide additional information to address any concerns.

---

### Official Review · Reviewer_LAee · 2023-08-12

**Soundness:** 3

**Excitement:**

3: Ambivalent: It has merits (e.g., it reports state-of-the-art results, the idea is nice), but there are key weaknesses (e.g., it describes incremental work), and it can significantly benefit from another round of revision. However, I won't object to accepting it if my co-reviewers champion it.

**Missing References:**

- Line 170 - missing citations for ATIS and Snips datasets
- Line 386 - missing citations for AGNews, IMDB, and Yahoo datasets
- Line 390 - missing citation for SST-2 dataset

**Paper Topic And Main Contributions:**

This paper proposes an augmentation scheme that takes the following steps. Given a set of source texts, (1) construct rules of grammar (RoG), which roughly corresponds to replacing NER tagged spans with placeholder slot variables like $artist, (2) alter the RoGs with one or more manipulation strategies to improve their contextual diversity, and finally (3) generate new sentences by sampling fill-values for the slots / contextual alternations. Experiments on several benchmark datasets show the benefits of GDA, especially for the lower resource settings and for datasets with more specific NER taggable tokens. Authors also went to great lengths to understand a key limitation of their technique, namely that it does not work on texts lacking NER taggable tokens.

Contributions include a new, well-performing augmentation pipeline that generates contextually diverse texts suitable for general NLP tasks. The analysis of why GDA fails to improve training for certain datasets represents a potentially useful diagnostic technique for future research in text augmentation, especially any techniques dependent on NER tagging / slot information.


**Questions For The Authors:**

- Question A: Can you elaborate on the details of how BERT and Bard were used to generate sentences? It would be helpful to understand Table 4 better, which shows an unexpected result where an older / smaller model like BERT outperforms a much larger and newer model like Bard on half the datasets.
- Question B: In Figure 6 and for SlotRefine on Snips, did you investigate the unexpected result of the 10-shot setting performing worse than the 5-shot setting for several techniques?
- Question C: What is the time efficiency of each step in the GDA algorithm?
- Question D: Are the training results robust (the average of multiple runs)?

**Reasons To Accept:**

- GDA explicitly considers both diversity and naturalness (syntactic / semantic fidelity) of the generations whereas many existing techniques yield ungrammatical, implausible, or incoherent texts. Some would argue that low-quality synthetic texts are acceptable because they still improve downstream model performance and generalization. However, in the real world, people often place significant weight on readability and having a technique that improves performance while aiming to be natural can help overcome objections from human decision makers and increase the likelihood of use.
- The evaluation setup acknowledges the limitations of the augmentation to low resource settings and sufficiently demonstrates its benefits relative to several existing baselines.


**Reasons To Reject:**

- Unclear why / how the authors base their approach on a context-free grammar when natural language is context-dependent. For example, in the sentiment analysis dataset IMDB, there may be a \\$director slot. It appears that GDA treats all directors as interchangeable even though there are a range of reputations that might influence the label semantics. “This movie was made by \\$director, so you know what to expect!” would likely entail positive sentiment if \\$director --> Steven Spielberg, likely negative if \\$director --> M. Night Shyamalan. In the conclusion, the paper claims that GDA “solves” issues with semantics, but it doesn’t seem like this can be true, at least not fully.
- In general, the claim of “solving” syntactic and semantic issues present in other augmentation techniques is left unquantified empirically, nor substantiated theoretically (likely not possible).
    - For syntax, authors could use a grammar checker like LanguageTool (https://github.com/languagetool-org/languagetool) to demonstrate that GDA does not introduce any new syntax issues.
    - For semantics, authors could use an approach like cleanlab (https://github.com/cleanlab/cleanlab) to demonstrate that GDA does not introduce any new label issues brought on by shifts in augmented semantics.

**Reproducibility:**

4: Could mostly reproduce the results, but there may be some variation because of sample variance or minor variations in their interpretation of the protocol or method.

**Reviewer Confidence:**

4: Quite sure. I tried to check the important points carefully. It's unlikely, though conceivable, that I missed something that should affect my ratings.

**Typos Grammar Style And Presentation Improvements:**

Typos:
- Line 196 - “... is context-free grammar” --> missing “a” before context-free

Presentation:
- Table 3 - recommend clarifying that the 5 / 10 row is for the shot-count.
- Lines 551-554 - the conclusions are stated rather strongly and without qualifications. For example, GDA appears to significantly worsen performance on datasets like Yahoo where there are not a lot of opportunities for slot annotation.

Difficult to Follow:
- Line 177 - the 3 phases of GDA do not match the ones described earlier on line 125. Recommend consistent terminology. Why does inference == RoG construction? Inference suggests a model is being used. Is it the NER tagging model? If so, how does GDA come up with the exact slot names like \\$artist and \\$song, which are more specific than person, place, and misc from NER tagging?
- Lines 196-232 - the terminology is not easy to follow and rule inference is unclear. For example, why do G and R share the same expression? Ultimately,
- Lines 400-402 - “These models…” is a confusing continuation from the previous sentence where the most recent items discussed are augmentation strategies.
- Line 468 - Unclear how IMDB is domain-unspecific since it exclusively features movie reviews and likely contains a lot of opportunities for slots like \\$actor, \\$movie, etc. In line 816, the paper appears to contradict itself by agreeing that IMDB is specialized.
- Lines 493-496 - Unclear how Bard and BERT were used in conjunction with GDA. Also it's a bit strange now to consider BERT an LLM with less than a billion params.
- Lines 823-827 - Unclear why a British corpus is being used to compare against datasets that tend to learn more towards American English. However, I still can see the value in performing this comparison to illustrate the point that Yahoo is more “generic” than other studied datasets. This point might have also been possible to make via perplexity or Salazar 2020’s LM scores, which are biased towards scoring general texts more highly.
    - Julian Salazar, Davis Liang, Toan Q. Nguyen, and Katrin Kirchhoff. 2020. Masked Language Model Scoring. In Proceedings of the 58th Annual Meeting of the Association for Computational Linguistics, pages 2699–2712, Online. Association for Computational Linguistics.

---

> ### Author Rebuttal · Authors · 2023-08-29
>
> We first thank the reviewer for the detailed comments and suggestions. We appreciate the productive remarks that would improve our paper. We promise to incorporate your feedback in our final version.
>
> **1. Details on the usage of context-free grammar**
>
> >Reasons to Reject #1-1)  Unclear why / how the authors base their approach on a context-free grammar when natural language is context-dependent. For example, in the sentiment analysis dataset IMDB, there may be a \\$director slot. It appears that GDA treats all directors as interchangeable even though there are a range of reputations that might influence the label semantics. “This movie was made by \\$director, so you know what to expect!” would likely entail positive sentiment if \\$director $\rightarrow$ Steven Spielberg, likely negative if \\$director $\rightarrow$ M. Night Shyamalan. In the conclusion, the paper claims that GDA “solves” issues with semantics, but it doesn’t seem like this can be true, at least not fully.
>
> **[Response]** We would first like to state that formal grammar, such as context-free grammar (CFG), is widely used in NLP, where numerous studies pay testament to its usefulness in various NLP tasks [1-5]. Context-free grammar is present in numerous facets of NLP, and in particular, is becoming more popular in neuro-symbolic approaches for interpretability. For example, Amazon recently published results and analyses of Alexa’s efficiency with regard to generating sentences based on formal grammar [6]. At the cross-section of natural and formal language, CFG, derived from several types of formal grammar, composes the underlying framework in which sentences are structured. Phrase structuring and sentence diagramming rules also [7]  utilize CFG to express natural language sentences. In [5], the researchers engage the self-training by using context-free grammar to pseudo-label the data and show meaningful performance improvements.
>
> The underlying rationale of GDA is based on rule sets derived from context-free grammar, but with the addition of slot information and rule manipulations, we are able to maintain the context of the original sentence. Our intention, which we stated in the abstract, but should have more thoroughly introduced in the methodologies section, is to make use of ‘context-aware grammar’, an extension of context-free grammar, as we are using the slot information as variables and restrict the grammar to generate sentences with semantic and syntactic similarity.
>
> >Reasons to Reject #1-2) For example, in the sentiment analysis dataset IMDB, there may be a \\$director slot. It appears that GDA treats all directors as interchangeable even though there are a range of reputations that might influence the label semantics. “This movie was made by \\$director, so you know what to expect!” would likely entail positive sentiment if \\$director $\rightarrow$ Steven Spielberg, likely negative if \\$director $\rightarrow$ M. Night Shyamalan. In the conclusion, the paper claims that GDA “solves” issues with semantics, but it doesn’t seem like this can be true, at least not fully.
>
> **[Response]** In regard to the influence of label semantics, this is an interesting and important facet of natural language, however we deemed it out of scope for our research as we aim to cover sentence semantics and syntax, but not necessarily sentiment. As we are dealing with data augmentation and producing sentences that are natural sounding and semantically and syntactically correct, we concluded that maintaining sentence sentiment does not impact the grammatical correctness of the augmented sentences. We do, however, find the idea of maintaining sentence sentiment while augmenting data to be intriguing and particularly interesting and would like to consider this as future work.
>
>
> [1] Valvoda et al., Benchmarking Compositionality with Formal Languages, Coling’22.
>
> [2] Igor Buzhinsky, Formalization of natural language requirements into temporal logics: a survey, INDIN’19.
>
> [3] Aydin et al., Domain Knowledge Representation Languages and Methods for Building Regulations, EBF’19.
>
> [4] Ziqi et al., Tab-CoT: Zero-shot Tabular Chain of Thought, ACL Findings’23.
>
> [5] Hahn et al., Self-training using rules of grammar for NLU, EMNLP Findings’21.
>
> [6] Tools for generating synthetic data helped bootstrap Alexa’s new-language releases: https://www.amazon.science/blog/tools-for-generating-synthetic-data-helped-bootstrap-alexas-new-language-releases.
>
> [7] An Intro to Phrase Structure Rules: https://www.linguisticsnetwork.com/an-intro-to-phrase-structure-rules.
>
> **2. Explanation of syntactic and semantic issues**
>
> >Reasons to Reject #2) In general, the claim of “solving” syntactic and semantic issues present in other augmentation techniques is left unquantified empirically, nor substantiated theoretically (likely not possible).
> For syntax, authors could use a grammar checker like LanguageTool (https://github.com/languagetool-org/languagetool) to demonstrate that GDA does not introduce any new syntax issues.
> For semantics, authors could use an approach like cleanlab (https://github.com/cleanlab/cleanlab) to demonstrate that GDA does not introduce any new label issues brought on by shifts in augmented semantics.
>
> **[Response]** The syntactic and semantic issues that we are claiming to solve relate to the possible logical errors (known syntactic and semantic issues) of current data augmentation (DA) techniques. Rather than focusing on logically correct sentence structure, we find that state-of-the-art DA methods use syntactic variation which begets logical errors. For instance, in Figure 1, ‘Add my playlist Bruno Mars Hits to a song by Bruno Mars’, generated by SSMBA, contains the logical error of adding a playlist to a song. Such errors can be potentially harmful to model performance, and thus are part of the impetus of our research
> On the other hand, GDA, based on CFG, restricts the structure of generated sentences through the usage of rules. For richer syntactic and semantic diversity, we also propose the rule manipulations depicted in Figure 3. Figure 2 also showcases the rules and manipulations which produce the diversified sentences generated by GDA.
>
> While it’s true that there is currently a tradeoff between focusing on sentence correctness and generating diverse sentences, we would like to affirm that GDA’s sentence correctness, especially grammatically, is competitive. However, the reviewers’ comments and proposed tools for analyzing the grammatical correctness of generated sentences are valuable. We’ll provide a detailed analysis of the grammatical accuracy in the final version.
>
> **3. Details on the experimental setting**
>
> &nbsp; **a.** How we used BERT and Bard to generate sentences
>
> >Question A) Can you elaborate on the details of how BERT and Bard were used to generate sentences? It would be helpful to understand Table 4 better, which shows an unexpected result where an older / smaller model like BERT outperforms a much larger and newer model like Bard on half the datasets.
>
> **[Response]** Given a 5-shot and a 10-shot dataset, we fine-tune BERT (stated in 4.2 last sentence) and then generate sentences with the fine-tuned BERT. For Bard, we designed a prompt that generates sentences when given a 5-shot or a 10-shot dataset. For instance when conducting 5-shot learning, we take 5 sentences sampled from the initial training set and ask Bard to generate semantically similar sentences.
> While BERT is fine-tuned with the training dataset, Bard is only used when prompted. Although Bard is the better-performing language model, fine-tuning plays a more critical role for generating more appropriate sentences - thus creating the performance difference between BERT and Bard.
>
> &nbsp; **b.** Details on the Snips dataset
>
> >Question B) In Figure 6 and for SlotRefine on Snips, did you investigate the unexpected result of the 10-shot setting performing worse than the 5-shot setting for several techniques?
>
> **[Response]** We presented our full experimental result in Appendix B regarding the combinations of our rule manipulations and hybrid strategies of language models and GDA. In Table 6, the result of SlotRefine on Snips dataset is questionable as 5-shot performance generally shows higher scores than the 10-shot performance.
> Unfortunately we did not investigate this unexpected result, however we can extrapolate from the data in Table 6 along with the dataset characteristics that the reason for this is that rule manipulation, as employed by GDA, generates syntactic and semantic variants of the original rules. These variants generally perform well, but variants of the Snips dataset induce noise that degrades the performance of SlotRefine. While the 10-shot performance of various combinations of GDA achieves lower than those of 5-shot, the initial grammar of 10-shot achieves the higher performance than that of 5-shot. Furthermore, for 10-shot, the performance of initial grammar shows a higher score than the performance of keyword, distance, and combined manipulations. Thus, for SlotRefine, rule manipulations on Snips dataset are less performant than the other datasets.
>
> &nbsp; **c.** Details on the learning efficiency and robustness
>
> >Question C) What is the time efficiency of each step in the GDA algorithm?
>
> >Question D) Are the training results robust (the average of multiple runs)?
>
> **[Response]** With regard to Question C - GDA works by 1) constructing an initial ruleset R from the given training dataset, 2) applying rule manipulations on R, and 3) generating sentences. The time complexity of the GDA algorithm depends on the number of sentences and their corresponding lengths. When we denote the number of sentences of a given dataset as N and the maximum length of sentences as L, the time complexity of the algorithm is O(NL). Thus, it takes a negligible amount of time to generate sentences by using GDA. We will update the time efficiency of GDA in the final version. If the reviewer still has open concerns regarding the time complexity of GDA, we are happy to provide a detailed explanation.
> With regard to Question D - We have multiple combinations of employing GDA depicted in Appendices A and B. We calculate the average scores of 5 runs for each combination.
>
> We appreciate your comments and eruditions, including new outlooks for evaluating syntactic and semantic correctness of GDA. We hope that our rebuttal clarifies your concerns of our approach in terms of its soundness. If you consider our responses inadequate, please let us know. We are delighted to supply you with the proper details for your concerns.

---

### Meta-Review · Program_Chairs · 2023-09-24

**Recommendation:** 3

**Metareview:**

This paper proposes a novel data augmentation technique for textual data, named GDA, which yields diverse and faithful samples. The methodology consists of inducing a grammar extended with slot information and diversifying it through rule manipulations. The results are especially promising in low-resource settings. Another strength of the paper is that it highlights its own limitations, as it mostly benefits tasks with NER-taggable tokens. However, the choice of CFG as the underlying grammar is questionable, as it restricts the space of sentences that can be successfully modelled by GDA. Another issue that remains not fully addressed is how to avoid error propagation in the proposed pipeline (e.g., incorrect NER slot predictions when these are not available in the data). To make the results more convincing, the Authors should compare the performance of DA baselines on datasets where slot annotation is available, possibly tone down their claims on reducing semantic errors which need more quantitative evidence, and add significance testing to the reported figures.

---

### Decision · Program_Chairs · 2023-10-07

**Decision:**

Accept-Findings

**Comment:**

This paper proposes a novel data augmentation technique for textual data, named GDA, which yields diverse and faithful samples. The methodology consists of inducing a grammar extended with slot information and diversifying it through rule manipulations. The results are especially promising in low-resource settings. Another strength of the paper is that it highlights its own limitations, as it mostly benefits tasks with NER-taggable tokens. However, the choice of CFG as the underlying grammar is questionable, as it restricts the space of sentences that can be successfully modelled by GDA. Another issue that remains not fully addressed is how to avoid error propagation in the proposed pipeline (e.g., incorrect NER slot predictions when these are not available in the data). To make the results more convincing, the Authors should compare the performance of DA baselines on datasets where slot annotation is available, possibly tone down their claims on reducing semantic errors which need more quantitative evidence, and add significance testing to the reported figures.